# On the characteristics of the wake of a wind turbine undergoing large motions caused by a floating structure: an insight based on experiments and multi-fidelity simulations from the OC6 Phase III Project

Stefano Cioni[1], Francesco Papi[1], Leonardo Pagamonci[1], Alessandro Bianchini[1*], Néstor Ramos-García[2], Georg Pirrung[2], Rémi Corniglion[3], Anaïs Lovera[4], Josean Galván[5], Ronan Boisard[6], Alessandro Fontanella[7], Paolo Schito[7], Alberto Zasso[7], Marco Belloli[7], Andrea Sanvito[8], Giacomo Persico[8], Lijun Zhang[9], Ye Li[9], Yarong Zhou[9], Simone Mancini[10], Koen Boorsma[10], Ricardo Amaral[11,12], Axelle Viré[12], Christian W. Schulz[13], Stefan Netzband[13], Rodrigo Soto-Valle[14], David Marten[15], Raquel Martín-San-Román[16], Pau Trubat[17], Climent Molins[18], Roger Bergua[19], Emmanuel Branlard[19], Jason Jonkman[19], Amy Robertson[19]

[1]Department of Industrial Engineering, University of Florence, Florence, 50139, Italy
[2]Department of Wind and Energy Systems, Technical University of Denmark, Lyngby, 2800, Denmark
[3]EDF R&D, Chatou, 78400, France
[4]EDF R&D, Palaiseau, 91120, France
[5]Department of Wind Energy, eureka!, Errigoiti, 48309, Spain
[6]Aerodynamic Department, Office National d'Etudes et de Recherches Aérospatiales, Paris, 92190, France
[7]Mechanical Engineering Department, Politecnico di Milano, Milano, 20156, Italy
[8]Laboratory of Fluid-Machines, Dipartimento di Energia, Politecnico di Milano, 20156, Italy
[9]Multi-function Towing Tank Laboratory, State Key Laboratory of Ocean Engineering, Shanghai Jiao Tong University, Shangai, 200240, China
[10]Wind Energy Department, Netherlands Organisation for Applied Scientific Research, Petten, 1755LE, The Netherlands
[11]Siemens Gamesa Renewable Energy, Saint Étienne de Rouvray, 76800, France
[12]Delft University of Technology, Faculty of Aerospace Engineering, Delft, 2629HS, The Netherlands
[13]Fluid Dynamics and Ship Theory, Hamburg University of Technology, Hamburg, 21073, Germany
[14]Department of Mechanical Engineering, Universidad de La Frontera, Temuco, 4811230, Chile
[15]Chair of Fluid Dynamics, Hermann-Föttinger-Institut, Technische Universität Berlin, 10623, Berlin, Germany
[16]Wind Energy Department, Centro Nacional de Energías Renovables (CENER), Sarriguen, 31621, Spain
[17]Nautical Science and Engineering Department, Universitat Politècnica de Catalunya, Barcelona, 08003, Spain
[18]Department of Civil and Environmental Engineering, Universitat Politècnica de Catalunya, Barcelona, 08034, Spain
[19]National Wind Technology Center, National Renewable Energy Laboratory, Golden, CO 80401, USA

*Correspondence to: Alessandro Bianchini (alessandro.bianchini@unifi.it)

**Abstract.** This study reports the results of the second round of analyses of the Offshore Code Comparison, Collaboration, Continued, with Correlation and unCertainty (OC6) project Phase III. While the first round investigated rotor aerodynamic loading, here, focus is given to the wake behavior of a floating wind turbine under large motion. Wind tunnel experimental data from the UNsteady Aerodynamics for FLOating Wind (UNAFLOW) project are compared with the results of simulations provided by participants with methods and codes of different levels of fidelity. The effect of platform motion both on the near and the far wake is investigated. More specifically, the behavior of tip vortices in the near wake is evaluated through multiple metrics, such as streamwise position, core radius, convection velocity, and circulation. Additionally, the onset of velocity oscillations in the far wake is analyzed because this can have a negative effect on stability and loading of downstream rotors. Results in the near wake for unsteady cases confirm that simulations and experiments tend to diverge from the expected linearized quasi-steady behavior when the rotor reduced frequency increases over 0.5. Additionally, differences across the simulations become significant, suggesting that further efforts are required to tune the currently available methodologies in order to correctly evaluate the aerodynamic response of a floating wind turbine in unsteady conditions. Regarding the far wake, it is seen that, in some conditions, numerical methods overpredict the impact of platform motion on the velocity fluctuations. Moreover, results suggest that the effect of platform motion on the far wake, differently from original expectations about a

faster wake recovery in a floating wind turbine, seems to be limited or even oriented to the generation of a wake less prone to dissipation.

# 1 Introduction

## 1.1 Background and motivation

Floating offshore wind turbines (FOWTs) have been one of the key study areas for wind energy research in the last few years because they represent the most promising way of exploiting the vast wind energy potential in deep waters. Despite recent research efforts, further work is required to improve the understanding of the complex interactions taking place between wind-driven loads, the aero-servo-elastic behavior of the rotor, and the hydrodynamics of the floater (Veers, 2022).

In particular, from an aerodynamic point of view, the effect of platform motion is two-fold. First, the flow field around the rotor is modified (Sebastian and Lackner, 2013; Tran and Kim, 2015; Chen et al., 2020), causing, for example, local differences in the relative wind speed. Second, during the complex motion of the platform, especially in case of severe sea states, the blades might enter their own wake, affecting the local induction and wake behavior (Dong and Viré, 2022; Ramos-García et al., 2022). All these effects might represent in theory a challenge for engineering models, which might no longer be applicable or require the introduction of additional corrections and tuning. For example, no agreement has been reached concerning the application of blade element momentum (BEM) methods to FOWTs when large amplitude of motions are considered (Tran and Kim, 2015; Farrugia et al., 2016; Ferreira et al., 2022).

The design of the turbine and floating substructure also requires an accurate evaluation of the hydrodynamic loads, induced by waves and sea currents (Gao et al., 2022). This represents a complex problem that depends on the specific floater architecture considered (Chen et al., 2020) and includes the coupling of wave excitation, potential flow radiation, and additional mass and viscous drag effects (Butterfield et al., 2007).

A further source of complexity is represented by the wave and wind forcing interaction. Floating wind turbines are a fully coupled system, where the motion of the platform affects the aerodynamic loading, and, at the same time, the wind loading has an impact on the position and velocity of the platform, and consequently on its hydrodynamic response (Chen et al., 2020). This results in a nonlinear response of the system to the combined external forcings, which limits the application of simplified linearized models.

In addition to external forcings, elasticity of both turbine and floater must be considered within the models to provide an accurate prediction of fatigue loading and improve the reliability of designs. Multiple aeroelastic models have already been proposed for fixed-bottom turbines, which can also be used in FOWTs.

Furthermore, the wind turbine controller has relevant effects on turbine loading and platform stability (Larsen and Hanson, 2007; Vanelli et al., 2022). One of the main issues concerns the platform pitch instability of FOWTs, which is caused by the coupling between the aerodynamic response of the rotor and the low-frequency fore-aft motion of the platform.

Hence, modeling approaches for FOWTs need to take into account and couple all the above-mentioned effects, resulting in a high level of complexity. Coupled areo-hydro-servo-elastic models have been developed (Jonkman and Matha, 2011; Chen et al., 2019; Ramos-García et al., 2021; Saverin et al., 2021), but accurate tuning and validation are required to evaluate their accuracy and reliability. Higher-fidelity approaches, such as computational fluid dynamics (CFD), are unfeasible at an early design stage due to the large computational cost, and their reliability still needs to be tested through accurate validation (Chen et al., 2020).

Among these different aspects, focus is given in this study to the turbine wake, the proper modeling of which is of paramount importance for future design of floating wind farms. For example, modeling the motion of the wake of a wind turbine, commonly referred to as wake meandering, affects both fatigue loading and power production of the downstream rotors (Larsen et al., 2008; Yang and Sotiropoulos, 2019). Another aspect of interest is the prediction of the distance at which the wind

velocity starts to recover. Nevertheless, the wake is characterized not only by a velocity deficit but also by the presence of multiple vortical structures, which affect the velocity recovery even after their collapse (Marten et al., 2020). In particular, the tip vortices form a coherent helical structure that dominates the near wake behavior before they merge and collapse in the far wake. Hence, understanding the mechanisms governing the tip vortices behavior can lead to better understanding of the wake dynamics.

Several researchers have provided insight into the behavior of the tip vortices shed from a fixed-bottom turbine; however, when moving to FOWTs, the analysis and modeling of wakes are further complicated by the effect of platform motion, which alters the aerodynamic response of the rotor. As a result, multiple differences can be observed between the wake of a fixed-bottom and a floating wind turbine. For example, Arabgolarcheh et al. (2022) used an actuator line model (ALM) CFD approach to show how surge and pitch motions of the platform affect the tip vortices by introducing periodic changes in their strength. This results in a highly unstable wake and a faster velocity recovery. Ramos-García et al. (2022) analyzed the wake shed from a FOWT under imposed surge and pitch motions, showing how the vortical structures in the near wake are modified for different frequencies of motion. Additionally, it was observed that the wake recovery is not greatly affected by the surge motion. Similarly, Tran et al. (2016) evaluated how the helical structure of the tip vortices is modified by surge motion through a CFD approach with the overset moving grid technique. Periodic changes in the spacing between subsequent tip vortices was observed, which could induce a faster collapse of the vortex structures.

General consensus supports the conclusion that the motion of FOWTs should aid mixing and recovery of the wake, which would be beneficial for the downstream turbines and allow a reduction of the spacing of the rotors. However, the platform motion could also induce low-frequency oscillations in the wake (Kleine et al., 2021), which could excite the response of the low-frequency platform modes (Veers et al., 2022). Wind tunnel measurements performed by Fontanella et al. (2022b) showed that surge and pitching motions of a FOWT can induce velocity oscillations in the wake that are propagated downstream at a characteristic speed. Additionally, CFD simulations performed by Kleine et al. (2022) using an Actuator Line Model showed that these perturbations can be amplified at specific frequencies of platform motion, resulting in large streamwise velocity oscillations.

**1.2 Scope of the study**

In the framework described above, the scarcity of open-access experimental data and validation tools represents one of the critical issues, currently hampering the progress of research (Veers et al., 2022). For this reason, the Offshore Code Comparison, Collaboration, Continued, with Correlation and uncertainty (OC6) Phase III project was developed under the International Energy Agency (IEA) Wind Technology Collaboration Programme Task 30 to validate rotor aerodynamic loading and wake behavior of a model FOWT. The OC6 participants, which include 28 universities and industrial partners from 10 countries, carried out simulations of a scaled version of the DTU 10 MW reference wind turbine and compared the results with the experimental data from the UNAFLOW campaign (Fontanella et al., 2021). The simulations were performed using a range of numerical methods with varying fidelity.

The analysis of the results from the OC6 Phase III project was divided into two parts. The first focused on the validation of rotor aerodynamic loading under large amplitudes of platform motion. This is a fundamental step because the aerodynamic response of the rotor is affected by the additional degrees of freedom of the system, and it is uncertain if the currently available simulation approaches developed for fixed-bottom turbines can provide a reliable estimate of the system loading. Indeed, even though fixed-bottom turbines are also affected by the oscillations of the rotor due to the elasticity of the system, the motion induced by the floating platform happens at lower frequency and higher amplitudes, complicating the adaptation of approaches developed for fixed-bottom turbines to floating systems. Bergua et al. (2023) compared thrust and torque oscillations from BEM, free vortex wake (FVW), and CFD simulations performed by the participants with the experimental data to evaluate the

capabilities of these approaches and identify possible issues and limitations of each methodology. The study was performed

over multiple load cases, providing further insight into the aerodynamic response of FOWTs.

In this work, which summarizes results of the second part of the OC6 Phase III project, focus is on the analysis of the fluid dynamics of the wake. To this end, only a subgroup of simulations was considered, i.e., those providing a solution of the wake itself. The methods used were FVW and CFD. FVW approaches employed in this study are based on the lifting line method. Hence, the wake is modeled by convecting the vorticity shed from a lifting line, where the circulation is calculated from

tabulated polars. On the other hand, CFD approaches solve the Navier–Stokes equations in the domain, providing better insight at the expense of a drastic increase in computational cost. A compromise between the two methodologies is represented by the ALM, which solves the fluid equations in the domain, except for the blade-flow interaction, which is replaced by sources of momentum in the corresponding mesh elements. These sources are determined from the same tabulated polars used by the FVW approaches.

More specifically, the aim of the study is to provide further insight into the modifications taking place in the near and far wake of a small-scale floating wind turbine under imposed surge and pitch motions by comparing experimental data with simulation results obtained with a range of methodologies by the participants. In this work, in order to simplify the description of the results, the near wake is defined as the region ranging from 0.25 to 0.5 diameters (D) downstream of the rotor, and the far wake is defined from 0.9D to 2.3D downstream.

In the near wake, the objective is to evaluate and quantify the effect of platform motion on the tip vortices by analyzing multiple tip vortex metrics, namely, position, core radius and strength. This is achieved by comparing velocity fields obtained through particle image velocimetry (PIV) with simulations in close proximity to the rotor. The goal is to evaluate from these results the aerodynamic response of a FOWT under imposed motion.

In the far wake (from 0.9D to 2.3D), the objective is instead to highlight the effect of platform motion on the wake recovery,

as this represents a crucial parameter for wind farm planning. Additionally, by comparing hot-wire anemometer (HWA) velocity data, the onset of velocity oscillations in the wake can be evaluated and quantified, providing valuable information that could be used to verify the stability and loading of downstream turbines.

This work is structured as follows. Section 2 provides a description of unsteady wake effects in wind turbine aerodynamics. The turbine model and experimental set up are presented in Sect. 3. Sections 4 and 5 describe the simulation approaches and

the test matrix, respectively. The postprocessing methodology is presented in Sect. 6, followed by the description of the PIV and HWA results in Sect. 7 and 8, respectively. Finally, conclusions and future work are discussed in Sect. 9.

## 2 Theoretical bases

The present section includes a brief overview of the modeling of the wake dynamics. This will be useful to interpret the results of the following sections.

**2.1 Linearized quasi-steady theory**

Floating horizontal-axis wind turbines are subjected to unsteady working conditions caused by platform motion. Understanding the aerodynamic response of the turbine to these unsteady conditions is crucial to predict both rotor loading and wake dissipation. The latter is strongly influenced by the behavior of the tip vortices, which is highly sensitive to the operating condition of the rotor.

A first approximation can be achieved by hypothesizing that the floating wind turbine operates in quasi-steady conditions, meaning that both loads and wake respond instantaneously to the platform motion; hence, the aerodynamic response of the turbine can be predicted from the fixed-bottom behavior. Previous works have shown that by applying a first-order

linearization, the rotor thrust and torque oscillations can be predicted under a range of amplitudes and frequencies of platform motion (Mancini et al., 2020; Fontanella et al., 2022a).

Following a similar approach, the tip vortex strength under platform motion can be described. Assuming that the circulation, $\Gamma$, of the tip vortex is a function of the average flow incidence in the last portion of the blade, which is in turn related to the angle of attack $\alpha$ at the blade tip,

$$\Gamma = \Gamma(\alpha) = \Gamma(U_{t,rel}, \omega_r), \tag{1}$$

and the circulation can be defined as a function of the rotational speed $\omega_r$ and tip relative velocity $U_{t,rel}$,

$$U_{t,rel} = (1 - a)U_\infty, \tag{2}$$

where $a$ is the induction factor and $U_\infty$ the freestream velocity. Hence, if the rotational speed is fixed and assuming that the induction factor is constant during the change of inflow velocity, the first-order linearization of the tip vortex strength is

$$\Gamma \approx \Gamma_o + \left.\frac{\partial \Gamma}{U_{t,rel}}\right|_0 (1 - a_0)(U_\infty - U_{\infty,0}), \tag{3}$$

where $(\cdot)0$ denotes the steady-state value of a quantity for a given turbine operating point. The difference in inflow velocity

between the fixed-bottom and imposed motion cases can be written as

$$(U_\infty - U_{\infty,0}) = -U_t, \tag{4}$$

where $U_t$ is the blade tip velocity in the streamwise direction. Considering either surge or pitching motions of the platform, and assuming small pitch angles, the streamwise position and velocity of the blade tip are defined as

$$x_t(t) = A_x sin(2\pi f_m t), \tag{5}$$

$$\dot{x}_t(t) = U_t = A_x 2\pi f_m sin\left(2\pi f_m t - \frac{\pi}{2}\right), \tag{6}$$

where $A_x$ is the blade tip streamwise motion amplitude, and $f_m$ the frequency of platform motion. Substituting Eq. (6) into Eqs. (3) and (4), the circulation can be expressed as

$$\Gamma \approx \Gamma_o - \left.\frac{\partial \Gamma}{\partial U}\right|_0 (1 - a_0)\left(A_x 2\pi f_m sin\left(2\pi f_m - \frac{\pi}{2}\right)\right), \tag{7}$$

which shows that, according to the linearized quasi-steady theory, the circulation of the tip vortices should oscillate around the

fixed-bottom value and be shifted by 90° compared to platform motion. The amplitude of the vortex strength oscillation (i.e., half of the maximum oscillation), normalized by $A_x$, can be calculated from the sinusoidal function of Eq. (7) as

$$\frac{\Delta \Gamma}{A_x} = K_\Gamma(a_0)(2\pi f_m), \tag{8}$$

$$K_\Gamma(a_0) = (1 - a_0)\left.\frac{\partial \Gamma}{\partial U}\right|_0. \tag{9}$$

If the first-order assumption is valid, and if the induction factor can be assumed constant, the oscillations of tip vortex strength

induced by the platform motion increase linearly with the frequency of motion. The slope of the curve is a function of the partial derivative of the circulation by the rotor relative velocity, calculated for the fixed-bottom case, and the steady-state induction factor.

Another crucial parameter when describing the wake of a wind turbine is the wake deficit (WD), which represents the velocity reduction due to momentum exchange with the wind turbine. The wake deficit at a fixed distance from the turbine is therefore

a function of the rotor relative velocity, $U_{rel}$, acting on the rotor and of the rotational velocity $\omega_r$,

$$WD = WD(U_{rel}, \omega_r), \tag{10}$$

and the same approach shown for the circulation can be replicated for the amplitude of the wake deficit oscillations:

$$K_{WD}(a_0) = (1 - a_0)\left.\frac{\partial WD}{\partial U}\right|_0, \tag{11}$$

$$\frac{\Delta WD}{A_x} = K_{WD}(a_0)(2\pi f_m). \tag{12}$$

Hence, if the induction factor is assumed constant, Eq. (12) represents the linearized quasi-steady response of the system. Following the same approach used for the tip vortex strength (Eq. (7)), the wake should be characterized by wake deficit oscillations that are shifted by 90° compared to platform motion. The amplitude of these oscillations increases with the frequency of platform motion (Eq. (11) and (12)), analogously to what was shown previously for the circulation of the tip vortices.

It is worth noting that other first order models (Johlas et al., 2021; Wei and Dabiri, 2022) have been proposed to predict the aerodynamic response of a FOWT under realistic floating motions. These models differ both in terms of amplitude and phase shift of the predicted rotor thrust and torque oscillations. For example, while Fontanella et al (2021), Mancini et al (2020), and Johlas et al (2021) predict that the rotor thrust should be in phase with the velocity profile, Wei and Dabiri (2022) have presented a model that shows a phase-shift between the rotor thrust and platform velocity in unsteady conditions. However,

different assumptions may lead to a different prediction of the aerodynamic response of a floating wind turbine. Analogously, assuming a different first-order linearization could lead to a different first-order model for wake deficit and tip vortex strength. The analysis of first-order models is out of the scope of this work, as they are used herein as a baseline comparison only.

### 2.2 Unsteady effects

As shown in the previous section, the linearized quasi-steady theory has been developed to predict the behavior of the wake

of a floating wind turbine under surge or pitch motions. However, this simplified theory does not take into account any unsteady effects.

Unsteady aerodynamic effects can be divided mainly into unsteady airfoil aerodynamic effects and dynamic inflow effects (Snel and Schepers, 1993; de Vaal et al., 2014). Most past studies have focused on the influence of these aerodynamic phenomena on rotor loads. At the airfoil level, the aerodynamic response of an oscillating profile can be described as a function

of the airfoil reduced frequency:

$$f_c = \frac{f_m c(r)}{2\sqrt{U_{rel}^2 + (r\omega_r)^2}}, \tag{13}$$

where $c$ is the blade chord, $U_\infty$ the freestream wind velocity. When the oscillation frequency increases, the airfoil response is no longer steady, meaning that the aerodynamic coefficients cannot be evaluated through the steady polars, using the time-

varying angle of attack. Sebastian and Lackner (2013) proposed an approximate threshold, Eq. (14), above which unsteady aerodynamic effects may become relevant:

$$f_{d,th} = \frac{0.05}{\pi c(r)} \sqrt{U_\infty^2 + (r\omega_r)^2}. \tag{14}$$

As it can be observed, the frequency threshold is more restrictive for the inboard parts of the rotor.

A further type of unsteady aerodynamic effects is dynamic inflow. Indeed, time variations of the vorticity trailed by the rotor affect the induced velocity and hence the aerodynamic behavior of the turbine. These variations can be caused by a change in rotor loading, which could be induced, for example, by the motion of the floating substructure or oscillations of the inflow velocity. Moreover, quasi-steady aerodynamic theories such as BEM assume that the inflow velocity is in equilibrium with the rotor loading, but in reality, a finite time is required for the induced velocity to reach the new equilibrium value, affecting

the distribution of the angle of attack on the blades. These effects can be parameterized through the rotor reduced frequency (Ferreira et al., 2022):

$$f_r = \frac{f_m D}{U_\infty}, \tag{15}$$

where $D$ is the rotor diameter. For an imposed surge motion of the turbine, experimental measurements performed by Fontanella et al. (2021) have shown that unsteady behavior in rotor thrust may be present when the rotor reduced frequency is larger than 0.5.

Generally, both unsteady aerodynamic response at the airfoil level and dynamic inflow effects will influence the rotor behavior. However, the two phenomena are characterized by different time scales (Snel and Schepers, 1993; Mancini et al., 2020). The airfoil-level effects have a shorter time scale in the order of the chord length divided by the relative velocity, while the dynamic inflow time constants are proportional to the rotor diameter divided by the wind speed. Previous works (Mancini et al., 2020; Fontanella et al., 2021) have characterized the aerodynamic response of a FOWT in terms of rotor loading as a function of the amplitude and reduced frequency as discussed in this section. In this work, the objective is to carry out an analogous investigation in terms of wake response.

## 3 Experimental data for benchmarking

Benchmarking experimental data refers to wind tunnel experiments carried out in the Politecnico di Milano's wind tunnel (GVPM) (Fontanella et al., 2021). A 1:75 scaled model of the DTU 10 MW reference wind turbine was tested both in a fixed-bottom configuration and with an imposed surge motion of the platform (i.e., translation of the platform in the direction of the wind). During a follow-on campaign, the model was tested for an imposed pitch motion; however, only force measurements were performed, and PIV and HWA data were not recorded. Figure 1 shows a simplified sketch of the turbine, including the definition of the main geometric features that are then summarized in Table 1. Further details about the turbine model and blade geometry are provided in the works by Bayati et al. (2018), Mancini et al. (2020), and Bergua et al. (2023).

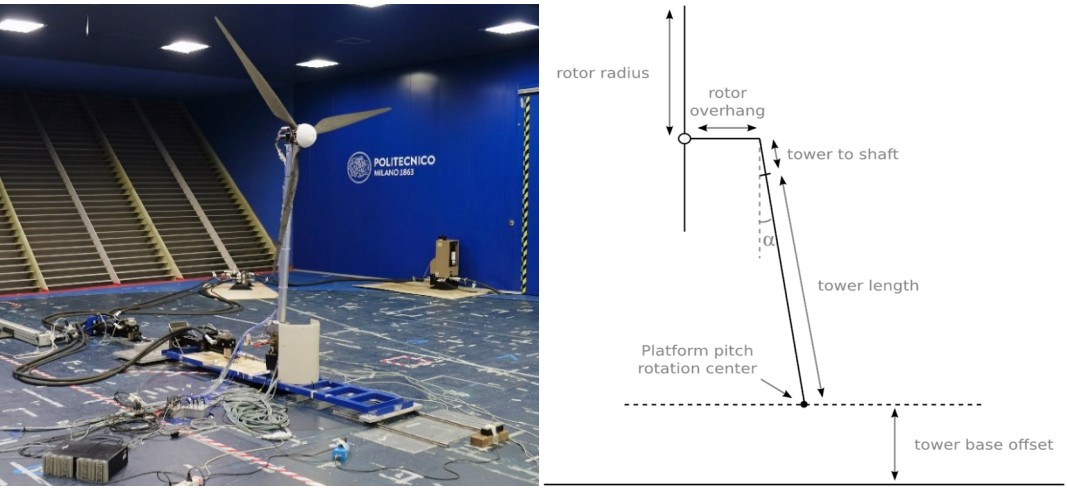

**Figure 1 Wind tunnel tests at Politecnico di Milano (UNAFLOW campaign) and sketch of the rotor geometry and reference system.**

**Table 1 Main geometrical parameters of the scaled wind turbine.**

| | |
|---|---|
| Rotor diameter (D) | 2.381 m |
| Blade length | 1.102 m |
| Hub diameter | 0.178 m |
| Rotor overhang | 0.139 m |
| Tower tilt angle ($\alpha$) | 5 deg |
| Tower to shaft distance | 0.064 m |
| Tower length | 1.400 m |

| Tower base offset | 0.730 m |
|---|---|

Multiple tests were performed during the experimental campaign, with varying inflow speed and tip speed ratio. During this work, only the tests cases at 4 m/s and a tip speed ratio of 7.5 were analyzed. Under these conditions, the Reynolds number is about $10^5$ for most of the blade span (Fontanella et al., 2021). The inlet turbulence intensity for the wind tunnel tests was 2% and the integral length scale was about 0.2 m (further details about the wind tunnel turbulence during the experimental campaign are reported in Appendix A).

The rotor diameter is equal to $D = 2.381$ m, resulting in a blockage ratio of about $\beta = 8\%$ (Robertson et al., 2023), estimated as the ratio of the rotor and wind tunnel test section areas:

$$\beta = \frac{\pi \left(\frac{D}{2}\right)^2}{WH} \tag{16}$$

where $W = 13.84\,m$ and $H = 3.84\,m$ are width and height of the wind tunnel test section. To account for blockage, some participants used a corrected inlet velocity, while the rest included the wind tunnel walls in their simulations. Further details about the methodology used by each participant are provided in Sect. 4. The corrected velocity was calculated using the correction proposed by Glauert for moderate blockage ratios (Inghels, 2013),

$$U'_\infty = U_\infty \left(1 + \frac{\beta C_t}{4\sqrt{1 - C_t}}\right)^{-1} \tag{17}$$

where $\beta$ is the area-based blockage ratio and $U'_\infty$ and $U_\infty$ are the corrected and actual free-stream velocities. The parameter $C_t$ is the thrust coefficient, calculated as:

$$C_t = \frac{T}{0.5\rho A_d U_\infty^2} \tag{18}$$

For the present case $C_t$ is about 0.88, $U_\infty$ is 4 m/s and the air density is 1.177 kg/m³ resulting in a corrected wind speed of 4.19 m/s.

During the experimental campaign, multiple measurements were carried out. Both PIV and HWAs were used to characterize the near and far wake, respectively. Additionally, the loading of the rotor was measured with a load cell at the tower top location, and the results have been used during the first part of the OC6 Phase III project to validate the simulation results in terms of rotor loading (Bergua et al., 2023).

The PIV measurements were carried out in the near wake of the rotor. The acquisition plane was vertical and positioned so as to capture the tip vortices that are shed by the wind turbine (Fig. 2). The acquisition of the velocity fields was carried out following two different strategies for the benchmark fixed-bottom case and for the surge cases.

Phase-locked velocity fields are acquired for the first case: the velocity is recorded when one blade, which is chosen as the reference, reaches a specific azimuth angle. During the experimental campaign, data from multiple revolutions are acquired and averaged in order to minimize the effect of measurement errors without losing information about the tip vortices. The available velocity fields are acquired for an azimuth position of the blade from 0° to 180° with a 15° interval and from 180° to 360° with a 30° interval. The velocity fields are used to track the tip vortices as they are convected downstream.

For the surge and pitch cases, the frequency of the platform motion was chosen as a multiple of the rotation frequency of the turbine; hence, the blades reach the same position over multiple cycles of the motion itself. PIV data were recorded over a cycle of platform motion, by recording the velocity fields for two fixed azimuth positions of the reference blade. Since PIV data were acquired only for a single cycle of platform motion for each load case, no averaging could be performed and the effect of cyclical dispersion on the results could not be evaluated. Further tests would be required in the future to be able to correctly validate the behavior of the tip vortices.

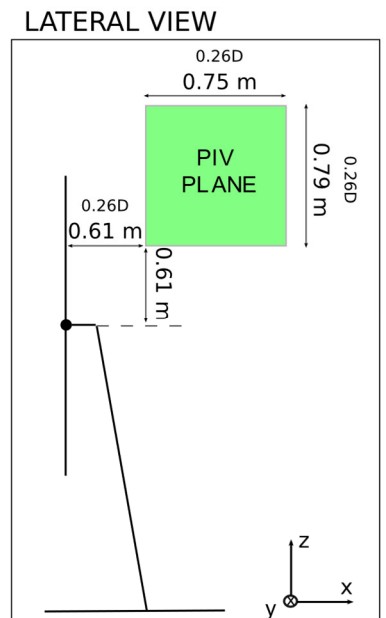
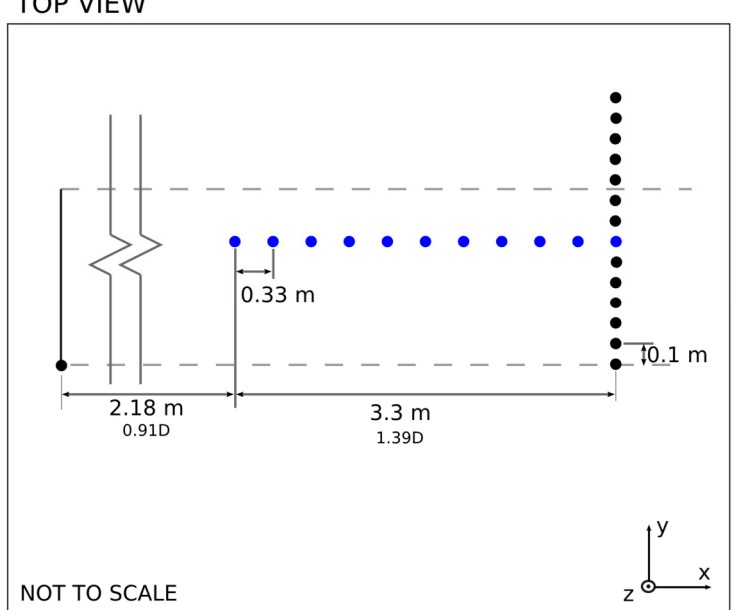

**Figure 2 PIV and HWA set up.**

Additionally, HWAs were used to characterize the behavior of the wake further downstream of the rotor. Two sets of measurements were carried out, one in the along-wind (x) direction and one in the crosswind (y) direction. The position of the probes relative to the rotor is shown in Fig. 2. Velocity data were acquired during the tests with an acquisition frequency of 2000 Hz. Further details about the experimental setup can be found in Fontanella et al. (2021).

## 4 Participants and numerical methods

The OC6 project involved contributions from 28 academic and industrial partners from 10 different countries. In this work the analysis is limited to the study of the wake; hence, only the results from participants that carried out FVW and CFD simulations are included. In total, 15 participants uploaded velocity data: Centro Nacional de Energías Renovables (CENER, Spain), Technical University of Denmark (DTU, Denmark), Électricité de France (EDF, France), eureka! (EURE, Spain), National Renewable Energy Laboratory (NREL, USA), Onera (ON, France), Politecnico di Milano (POLIMI, Italy), Shanghai Jiao Tong University (SJTU, China), Netherlands Organization for Applied Scientific Research (TNO, Netherlands), Technical University of Berlin (TUB, Germany), Hamburg University of Technology (TUHH, Germany), Technical University of Delft (TUD, Netherlands), Università degli studi di Firenze (UNIFI, Italy), and Universitat Politècnica de Catalunya (UPC, Spain). A summary of the participants and methodologies is provided in Table 2. DTU performed both FVW and CFD simulations, while the majority focused on a single approach. Both FVW and CFD simulations were set up to replicate the experimental campaign. The results were obtained after multiple rotor revolutions and cycles of platform motion to guarantee convergence of the results.

For the FVW methods, lift and drag coefficients for 20 radial stations were provided to the participants. These include seven sets of coefficients over a range of Reynolds numbers between $5 \cdot 10^4$ and $5 \cdot 10^5$ (Robertson et al., 2023). The FVW participants can be divided into two groups: ON and TUHH used the provided polars as a look-up table (static polars approach), while the remaining participants accounted for unsteady airfoil aerodynamics. Indeed, dynamic changes in the inflow velocity can lead to hysteresis in the airfoil behavior, both during attached flow and stall conditions (Theodorsen, 1949; Leishman, 2006). The codes using static polars can capture the circulatory unsteady aerodynamic effects (i.e., the hysteresis in attached flow), while dynamic stall is only captured by the remaining participants. These unsteady effects are evaluated by the simulation code used with a methodology that depends on the applied numerical approach.

**Table 2 List of participants and modeling approaches and codes.**

| Participant | Code Name | Underlying aerodynamic modeling | | | |
|---|---|---|---|---|---|
| | | FVW | CFD | | |
| | | | HYBRID PARTICLE-MESH | ACTUATOR LINE MODEL | BLADE RESOLVED |
| **CENER** | AeroView | ✓ | | | |
| **DTU** | HAWC2-MIRAS | ✓ | ✓ | | |
| **EDF** | DIEGO | ✓ | | | |
| **EURE** | OpenFAST | ✓ | | | |
| **NREL** | OpenFAST | ✓ | | | |
| **ON** | PUMA | ✓ | | | |
| **POLIMI** | Modified OpenFOAM | | | ✓ | |
| **SJTU** | STAR-CCM+ | | | | ✓ |
| **TNO** | AeroModule | ✓ | | | |
| **TUB** | QBlade | ✓ | | | |
| **TUHH** | panMARE | ✓ | | | |
| **TUD** | YALES2 | | | ✓ | |
| **UNIFI** | CONVERGE | | | ✓ | |
| **UPC** | OpenFAST | ✓ | | | |

Additionally, FVW methods can be distinguished in terms of the methodology used to simulate the wake (Table 3). For example, some included a core radius growth model, or imposed the initial core size of the tip vortices following different approaches. Despite the differences in the simulation setup, EURE, NREL, and UPC used the cOnvecting LAgrangian Filaments (OLAF) code to perform the FVW simulations (Shaler et al., 2020; Branlard et al., 2022). While CENER used Aerodynamic Vortex fIlamEnt Wake (AeroVIEW) (Martín-San-Román et al., 2021). DTU performed the simulations using the solver HAWC2-MIRAS (Ramos-García et al., 2021). EDF employed a modified version of OLAF (Corniglion, 2022). TNO used the Aerodynamic Windturbine Simulation Module (AWSM) (Van Garrel, 2003). TUHH carried out the simulations using the LL extension (Wang and Abdel-Maksoud, 2020) of the panel method panMARE (Netzband et al., 2018).

Among CFD participants, POLIMI and UNIFI performed unsteady Reynolds-averaged Navier–Stokes (RANS) simulations using an ALM approach. POLIMI and UNIFI used two different turbulence models: the former used the k-$\omega$ SST model while the latter used the k-$\varepsilon$ RNG model. SJTU carried out blade-resolved simulations with a body-fitted grid method and by using the Spalart-Allmaras turbulence model. A large-eddy simulation (LES) ALM approach with the dynamic Smagorinsky model was used by TUD. Even though the turbulence model used in the simulations may affect the results, a sensitivity analysis to different turbulence models was not performed as it was out of the scope of this work. Despite the different methodologies employed, the participants used comparable element sizes to perform the spatial discretization in the rotor and near wake region, including the PIV plane (about $10^{-2}$ m). Moving away from the rotor, the participants using RANS approaches (POLIMI, UNIFI, and SJTU) increased the element size to reduce the computational cost. In contrast, TUD maintained the

same refinement up to 2.5 diameters from the rotor while performing LES simulations. Further details about the mesh sizing and methodology employed by the CFD participants are summarized in Table 4. The participants tested different mesh sizes to evaluate the effect of the grid sizing on the results and to guarantee accuracy. All CFD participants, except DTU and SJTU, included the walls in their simulations; hence, no correction of the freestream velocity was performed to account for the blockage. In detail, POLIMI and UNIFI applied a slip wall condition to the boundaries and reduced the size of the numerical domain by the boundary layer displacement thickness. On the other hand, TUD did not decrease the size of the domain and solved the boundary layer.

The ALM simulations (POLIMI, TUD, and UNIFI) used the steady polars provided to the participants. Among the CFD participants, DTU used a hybrid particle-mesh method, which accounts for vortex diffusion and stretching by solving the Navier–Stokes equations.

Table 3 Main simulation parameters for FVW simulations. Default* indicates that the participant used the default value available within the numerical code.

| | Vortex element type | Blade discretization | Biot–Savart Kernel | Initial core radius | Core growth model | Wake length |
|---|---|---|---|---|---|---|
| CENER | Filaments | 20 elements | Vatistas core model | $0.05c$ | Yes | 3.75D |
| DTU | Filaments and mesh | 40 elements | 10th-order Gaussian | Not defined | No | 10.71 m |
| EDF | Filaments | 40 elements | Vatistas core model | $0.01c$ | No | ~11D |
| EURE | Filaments | 20 elements | Vatistas core model | $0.25\delta r$ for wing regularization $0.5\delta r$ for wake regularization | No | 5D |
| NREL | Filaments | 20 elements | Vatistas core model | $0.05c$ for wing regularization $0.3c$ for wake regularization | Yes | ~4.5D |
| ON | Vortex sheet | 45 elements | Default* | Default* | Default* | 25 rotor revolutions |
| TNO | Filaments | 20 elements | Smooth core model | 1 % of filaments length for induced velocities on blade lifting line 20 % of filaments length for wake velocities | No | 3D |
| TUB | Filaments | 30 elements | Van Garrel type core model | $0.05c$ | Yes | 17 rotor revolutions |
| TUHH | Dipole panels | 25 elements | Lamb–Oseen core model | 0.071 m | No | ~6D |
| UPC | Filaments | 20 elements | Vatistas core model | $0.6\delta r$ for wing and wake regularization | No | ~3D |

**Table 4 Main simulation parameters for CFD simulations**

| Participant | POLIMI | SJTU | TUD | UNIFI |
|---|---|---|---|---|
| Simulation approach | ALM URANS | Blade resolved DES | ALM LES | ALM URANS |
| Turbulence model | k-$\omega$ SST | Spalart-Allmaras | Dynamic Smagorinsky | k-$\varepsilon$ RNG |
| Rotor region [D] | 0.26 | 0.11 | 2.5 | 0.22 |
| Rotor region cell size | $1.7 \cdot 10^{-2} m$ | $3 \cdot 10^{-3} m$ | $1.3 - 2.4 \cdot 10^{-2} m$ | $1.56 \cdot 10^{-2} m$ |
| Near wake region [D] | 0.63 | 6.26 | N.A. | 0.84 |
| Near wake element size | $2 \cdot 10^{-2} m$ | $1.2 \cdot 10^{-2} m$ | N.A. | $3.13 \cdot 10^{-2} m$ |
| Far wake element size | $4.5 \cdot 10^{-2} m$ | $4.8 \cdot 10^{-2} m$ | $5 \cdot 10^{-2}$ | $6.25 \cdot 10^{-2} m$ |

## 5 Load cases under investigation

Multiple load cases (LCs) of platform motion are analyzed (see Table 4). The fixed-bottom results (LC1.1) are postprocessed to provide a benchmark for the LCs characterized by the motion of the platform. For this LC, experimental PIV and HWA data are available. The participants postprocessed the results in order to achieve analogous outputs to the experimental measurements. Hence, for the PIV data, the outputs are the velocity fields captured for the same azimuth positions of the reference blade (see Sect. 3), and for the HWA the outputs are the velocity values at the coordinates of the probes. For all load cases, the wind velocity was equal to 4 m/s (before accounting for blockage) with a turbulence intensity of 2%, and the rotational velocity of the turbine was set to 4 Hz (240 rpm) (Bayati et al., 2018). The numerical models did not account for wind turbulence and simulated a steady inflow.

Unsteady cases include surge and pitch motions of the platform. For both cases, a purely sinusoidal motion is considered. The motion follows a negative sinusoid for the surge cases (i.e., the platform moves upstream at the beginning of the cycle) while for the pitch cases a positive sinusoidal is used (i.e., the platform moves first downstream). Selected amplitudes and frequencies of motion are considered representative of a floating horizontal-axis wind turbine (Mancini et al., 2020). The dimensions of the turbine were scaled by 75, while the wind speed was scaled by a factor of 3. The scaling was performed keeping the rotor reduced frequency and the normalized amplitude of motion, defined as A/D for the surge cases and as the angle of motion for the pitch cases, constant (Mancini et al., 2020). In this way, the relationship between the wind and platform velocity is preserved.

Experimental PIV and HWA data are available for the surge cases only. A second experimental campaign was carried out by POLIMI, which included only force measurements for a pitch motion of the platform; the results are included in the first part of the OC6 project (Bergua et al., 2023). Since the participants had already run simulations for the pitch cases, the wake results for these additional load cases are included here in order to compare surge and pitch results. Additionally, the differences between multiple simulation approaches can be underlined for a pitch motion of the platform.

**Table 5 Summary of analyzed load cases and corresponding amplitudes and frequencies of motion. The values of reduced frequency are calculated from Eq. (15), using the corrected free-stream velocity to account for wind tunnel blockage. Check marks represent whether experimental or simulation data were available during the project.**

| Platform Motion | Load Case | Platform Amplitude [m] or [deg] | Platform Frequency [Hz] | Reduced Frequency | Experimental Data | Simulation Data |
|---|---|---|---|---|---|---|
| **Fixed-bottom** | 1.1 | None | | 0 | ✓ | ✓ |
| **Surge** | 2.1 | 0.125 | 0.125 | 0.071 | ✓ | ✓ |
| | 2.5 | 0.035 | 1.000 | 0.568 | ✓ | ✓ |
| | 2.7 | 0.008 | 2.000 | 1.137 | ✓ | ✓ |
| **Pitch** | 3.1 | 3.0 | 0.125 | 0.071 | | ✓ |
| | 3.5 | 1.4 | 1.000 | 0.568 | | ✓ |
| | 3.7 | 0.3 | 2.000 | 1.137 | | ✓ |

## 6 Data postprocessing

In this section, the postprocessing methodology is described. Both PIV velocity fields and HWA data were analyzed for all the available LCs described in Sect. 5.

### 6.1 PIV postprocessing

The analysis of PIV data was focused on the behavior of the tip vortices under platform motion conditions. As described in Sect. 3, two different acquisition strategies have been used for the fixed-bottom case (LC1.1) and for the platform motion cases. During LC1.1, phase-locked velocity field data were acquired over a range of 360° of the reference blade. This means that multiple velocity snapshots are available where the tip vortex is captured at different time steps while it is convected downstream. Hence, the evolution of the tip vortex can be described. The most common approach in the literature is to describe the time discretization in terms of vortex age (Soto-Valle et al., 2020): the time that passes between the vortex shedding and the analysis of the tip vortex is defined in terms of the azimuth angle traveled by the blade (see Fig. 3).

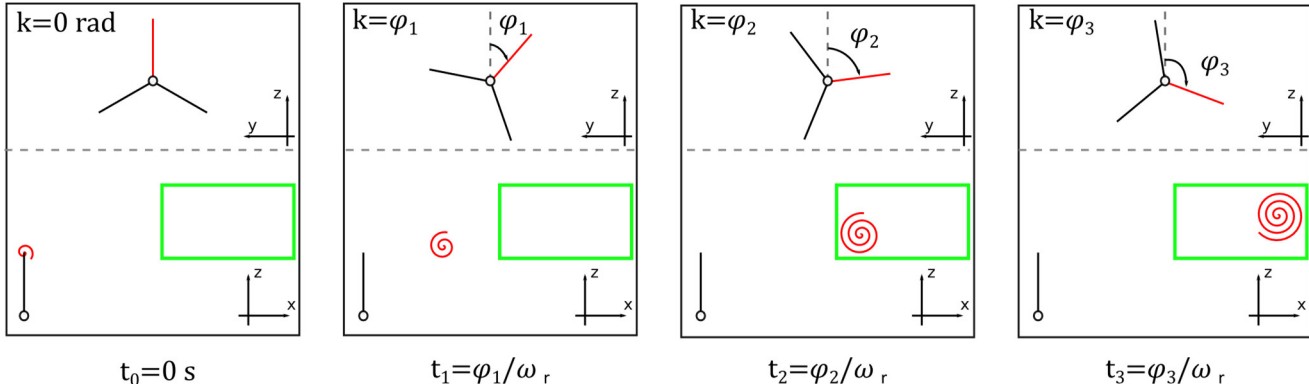

**Figure 3 Sketch of the definition of vortex age. The time that passes from the formation of the tip vortex to the time it is analyzed is expressed in terms of the azimuth angle of the blade.**

For the platform motion cases, the velocity fields were acquired only for two azimuth positions of the blade during a cycle of platform motion. Thus, the velocity fields only capture the tip vortex for a limited number of vortex ages and the analysis was focused on evaluating the effect of surge and pitch motion on the tip vortex parameters described in this section.

The first step of postprocessing concerns the identification of the tip vortices from velocity data. Multiple methods have been proposed in the literature to perform this step. Soto-Valle et al. (2022) have shown that the most suitable approach is Graftieaux's method (Graftieaux et al., 2001), which identifies the tip vortices from the velocity field without needing to

evaluate the velocity gradients. The tip vortices are identified from the local maxima of the scalar $G_1$, defined by Graftieaux's method.

The tip vortex location can be used to track the tip vortices during their convection. Then, the location of the vortex center is used to calculate further metrics such as core radius and convection velocity. When analyzing vortex structures, the vortex core is generally defined as the inner part of the structure, where the fluid rotates as a rigid body (van der Wall and Richard, 2006). Hence, this region is characterized by high vorticity. In this work, the core radius is estimated from horizontal slices of the velocity field, passing through the vortex center. Half the distance between the two maxima of the swirling velocity profile is assumed equal to the core radius (Soto-Valle et al., 2022).

The convection velocity can be estimated only for the fixed-bottom cases, due to the limited number of vortex ages available for the unsteady cases. The convection velocity is calculated from the tip vortex trajectory by estimating the slope of the streamwise position and time curve.

The tip vortex strength is calculated by integrating the vorticity distribution, $\omega$, around the vortex center:

$$\Gamma = \int_A \omega dA \tag{19}$$

After an accurate analysis of the vorticity distributions of both simulations and experiments, the integration area was chosen as a 125 x 75 mm$^2$ region. The integration surface should include all the vorticity associated with the tip vortex but also exclude unwanted contributions. For both the experiment and CFD simulations that included the wind tunnel walls, the tip vortex is found further inboard than expected due to wind tunnel blockage, as observed in the first part of this work (Bergua et al., 2023). Hence, the tip vortex is found closer to the vorticity shed from the inner parts of the blade, and the integration surface was limited in the radial direction to 75 mm to avoid the inclusion of these vorticity contributions (see Fig. 4).

Additionally, CFD results showed further spreading of the vorticity distribution in the streamwise direction, therefore the integration domain was extended to 125 mm to include all the vorticity contributions and at the same time avoid the inclusion of vorticity from the previous or following vortices found in the field of view. The same integration domain was used for all the participants and modeling approaches, excluding the simulation data by TUB, which in contrast to the other results showed further vorticity spreading in the radial direction compared to the streamwise direction. To account for this, the integration area was extended in the radial direction to 125 mm when postprocessing the results obtained by TUB. Calculating the circulation from Eq. (19) may lead to some errors due to the evaluation of the velocity derivatives using a finite differencing scheme, which introduces a truncation error and could amplify measurement errors for the experimental data. The error introduced by the methodology used in this work was estimated as less than 1.5% for all numerical and experimental results (see Appendix A for further details on uncertainty on tip vortex circulation estimation).

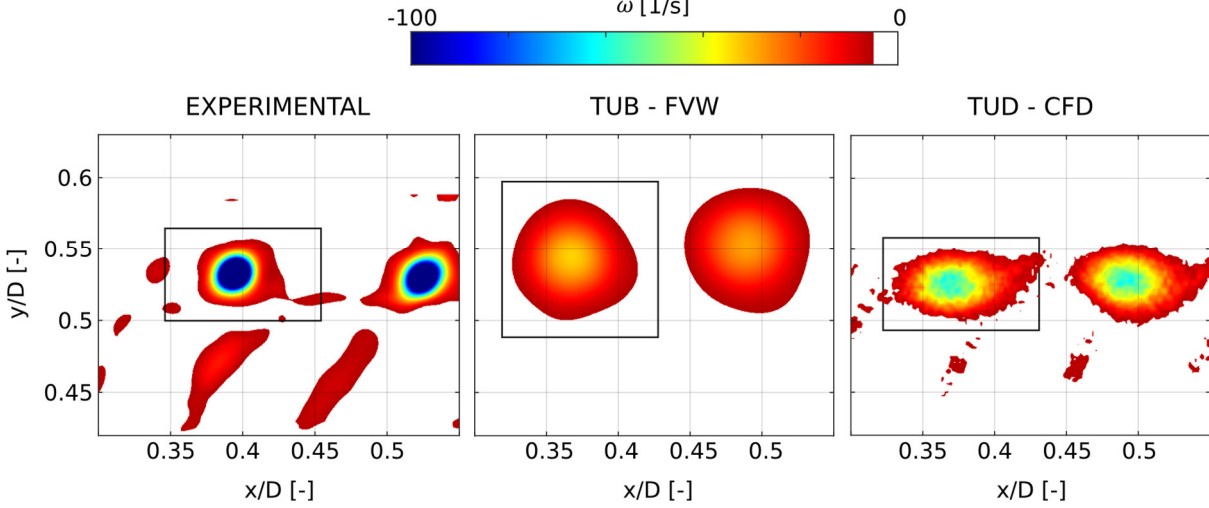

**Figure 4 Sketch of integration area used to evaluate the circulation. Using a rectangular integration area allows the inclusion of all the vorticity contributions from the tip vortex without including the vorticity associated with the blade passage. The integration area is expanded in the radial direction when the data from TUB is analyzed in order to account for the spreading of vorticity in the radial direction. The vorticity plots are shown using a threshold of $\omega = 5\ \mathrm{m^2/s}$.**

For unsteady cases, the effect of platform motion on the tip vortices can be evaluated by analyzing how the discussed tip vortex properties change during a cycle of platform motion. The simulations showed a linear response to the imposed platform motion, as the position and strength of the tip vortices oscillate following a sinusoidal at the same frequency. For this reason, the data were analyzed by applying the Fourier transform to the vortex metrics results and by evaluating the amplitudes and phase shifts of the signals at the frequency of platform motion (Fig. 5). In order to evaluate the phase shift regarding comparison to platform motion, results must be shifted in the time domain, in order to account for the time required for the tip vortex to reach the field of view. Since the velocity fields are acquired for specific azimuth angles of a blade, which is chosen as reference, the tip vortex age is known, and the time required by the tip vortex to enter the field of view can be calculated as

$$t = \frac{K}{f_t \cdot 360°},$$ (20)

where $K$ is the vortex age (expressed in degrees) and $f_t$ is the rotational frequency of the rotor.

The results in terms of amplitude and phase shift from both the experimental campaign and simulations can then be compared. For the experimental data, the fast Fourier transform can be applied only to LCs 2.1 and 2.5 due to the limited amount of velocity fields available for LC2.7. For the simulations, the PIV data were recorded following the same strategy as the experimental campaign for LCs 2.1 and 3.1, but the number of velocity fields was increased for the remaining load cases by acquiring the velocity fields from the three blades and not only one. In this way, the behavior of the tip vortices during surge and pitch motions can be analyzed even for those LCs where, due to the high frequency of platform motion, the number of available PIV velocity fields during a cycle of platform motion is limited (four velocity fields for LCs 2.5 and 3.5, and two for LCs 2.7 and 3.7).

To compare experimental data and numerical results, the uncertainty of the tip vortex metrics needs to be estimated. For the fixed-bottom case, the post processing was performed on the phase-locked velocity fields obtained by averaging the velocities over 100 rotor revolutions. Hence, in this case, the standard deviation of all the metrics was estimated by replicating the analysis for all the velocity fields. However, the same methodology could not be applied to the surge cases, as the phase-locked velocity fields were acquired experimentally for a single cycle of surge motion, due to limitations in the experimental wind tunnel availability. Therefore, the uncertainty of the tip vortex metrics for the surge cases could not be estimated.

## 6.2 HWA postprocessing

Two sets of data are available from HWA measurements, i.e., velocities from the along-wind probes and those from the crosswind ones (see Fig. 2). The along-wind probes can be used to evaluate the velocity trends when increasing the distance from the rotor. When a platform motion is imposed, a streamwise velocity perturbation is propagated in the wake. The velocity oscillation can be analyzed for each along-wind probe by performing a Fourier transform of the streamwise velocity signal. The amplitude at the frequency of platform motion can then be extracted. However, the phase shift of the velocity oscillations with regard to platform motion cannot be evaluated, since the convection velocity of the velocity perturbation changes with the distance from the rotor (Fontanella et al., 2022b).

The crosswind probes can be used to evaluate the wake deficit caused by the extraction of energy from the turbine. The wake deficit for LC1.1 was already evaluated in the first part of the OC6 project (Bergua et al., 2023). The same definition of the wake deficit is used here, based on spatial averaging:

$$WD = \frac{\sum_{i=1}^{N} |r_i| \cdot (u(r_i) - u_\infty)}{\sum_{i=1}^{N} |r_i|},$$ (21)

where $r$ is the radial coordinate and $u$ is the streamwise velocity. The wake deficit is weighted by the radial coordinate, providing a rotor averaged value. Additionally, the radial scaling reduces the impact of the differences between experimental and simulation data observed near the hub. Indeed, all participants, except UNIFI, did not model the blockage due to rotor hub

and nacelle nose, and this leads to a reduced velocity deficit at small radial coordinates ($r < 0.2$ m), as shown by Bergua et al. (2023).

In the present study, the effect of platform motion is evaluated by calculating the wake deficit during a cycle of surge or pitch motion. The average wake deficit can be evaluated and compared with the fixed-bottom case. Additionally, the wake deficit trend is characterized by oscillations induced by the motion. This effect can be quantified by applying the Fourier transform

to the wake deficit results to extract the amplitude of the oscillations at the frequency of motion.

Hotwire measurements were performed over multiple surge cycles. Hence, to aid the comparison of numerical data with the experimental results, the standard deviation of the streamwise velocity and wake deficit was calculated from the available data. However, the uncertainty connected to the amplitude and phase-shift of the streamwise velocity and wake deficit could not be evaluated due to the limited number of surge cycles available.

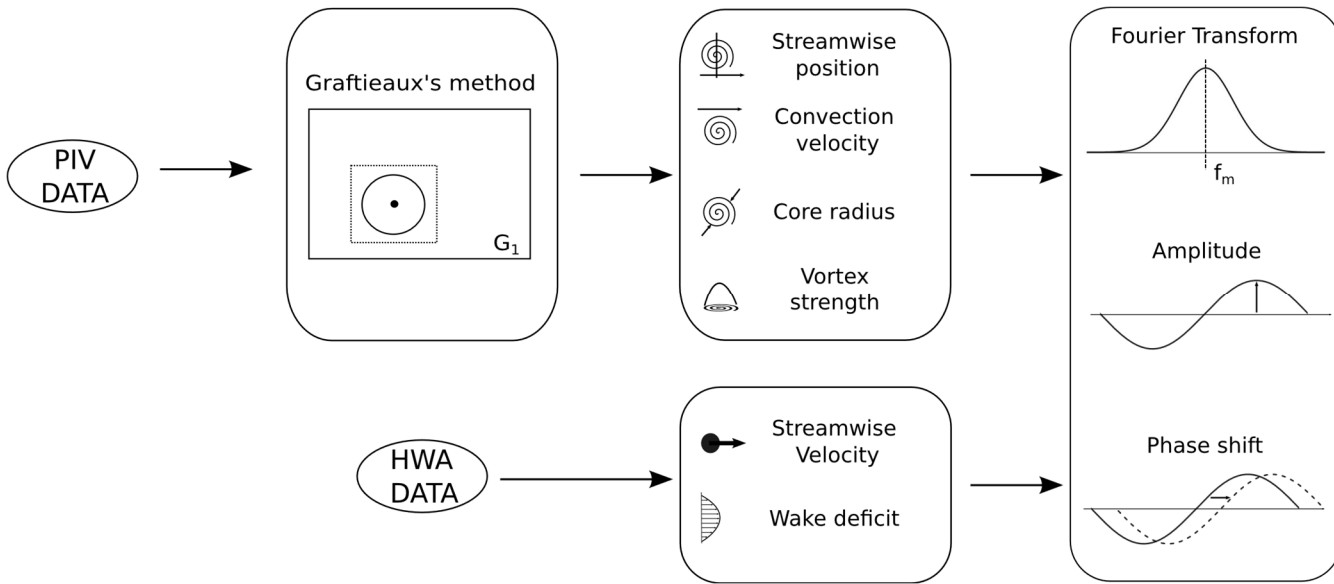

**Figure 5 Summary of postprocessing.**

## 7 PIV results

The results from the analysis of the PIV velocity fields are presented in this section. Section 7.1 shows the results from the

500 fixed-bottom case to provide a benchmark for the platform motion cases. Then, surge and pitch results are described in Sect. 7.2.

### 7.1 Fixed-bottom results

For the fixed-bottom case, LC1.1, the velocity fields were acquired for multiple vortex ages (see Sect. 3). The behavior of the tip vortex can then be described as a function of the vortex age. The parameters of interest are the streamwise position,

convection velocity, core radius, and strength. In the following figures the dashed lines represent the FVW results while the solid lines represent either the CFD or experimental results. Additionally, error bars show the standard deviation of experimental data, calculated from the velocity fields captured for 100 rotor revolutions (see Sect. 6.1 for further details).

Figure 6 (left) shows the tracking of the tip vortex in the streamwise direction, as it is convected downstream. The tip vortex streamwise position increases linearly with the vortex age, meaning that the convection velocity of the tip vortex is constant

in the near wake, in agreement with previous wind tunnel experiments (Snel et al., 2007; Yang et al., 2012; Ostovan et al., 2018; Soto-Valle et al., 2020). The slope of the curve is a representation of the streamwise convection velocity (Fig. 6). The results obtained through FVW or CFD simulations show maximum differences of about 0.03D in terms of the streamwise position of the tip vortex.

Comparing the convection velocity (Fig. 6, right) from all simulation approaches and the experimental results, it can be observed that only NREL overpredicts the convection velocity by about 3 %. Instead, the convection velocity is smaller for the rest of the participants by 1 % to 10 %, both for those that included the wind tunnel walls and for those that did not. The convection velocity of the tip vortices is a function of the operating conditions of the wind turbine. In this case, the experimental tip speed ratio is $\lambda = \frac{\omega R}{U_\infty} = 7.5$, the thrust coefficient is $C_T = 0.87$ (Fontanella et al., 2022b), and the axial induction is $a = 0.32$.

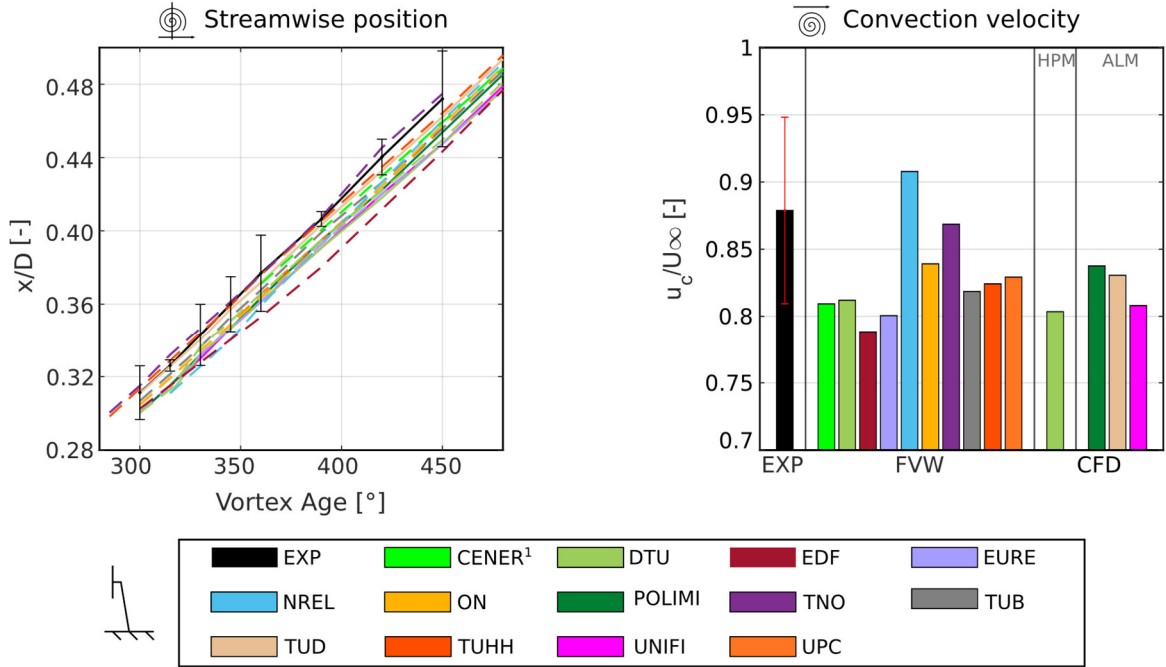

**Figure 6 Tip vortex streamwise position (left) and convection velocity (right). Dashed lines represent the FVW results and solid lines represent either the CFD or experimental results. [1]Error bars represent the standard deviation of the experimental data.**

The simulations predict a convection velocity that is about 85 % of the freestream velocity, in line with previous experimental campaigns and simulations for fixed-bottom turbines (Snel and Schepers, 1993; Soto-Valle et al., 2020). Additionally, the convection velocity is commonly approximated as a function of the axial induction and freestream velocity (Okulov and Sørensen, 2010; Pirrung and Madsen, 2018; Corniglion et al., 2022), as $u_c = (1 - k_c a)U_\infty$. Previous experiments have shown that the constant $k_c$ ranges from 0.5 to 1.5 (Odemark and Fransson, 2013; Boorsma et al., 2018). In this case, the value of $k_c = 0.5$, calculated from the induction factor and freestream velocities reported above, provides a good approximation for both experiments and simulations. The behavior of tip vortices is also described in terms of vortex core radius and tip vortex strength (Fig. 7). In terms of vortex core radius, some differences are observed across the participants. The experimental results show that the core radius $r_c$ increases linearly with vortex age from a value of about 2.2 % of the rotor radius $R$ to about 2.6 %, similar to what was observed in previous experiments concerning horizontal-axis wind turbines (Ebert and Wood, 1999; Massouh and Dobrev, 2014; Soto-Valle et al., 2020). Among the FVW results large differences are found, probably due to different implementations in the simulation codes that were tested. Indeed, FVW approaches do not solve the tip vortex but model the behavior of the core. For example, the initial core radius is imposed using various methodologies, such as a function of the chord of the last section of the blade (CENER and TUB, imposed $r_c = 5$ %) or as a function of the filament length (TNO). Additionally, some codes do not include a vortex core growth model (such as EURE, DTU, TUHH, and TNO), while others do (CENER, NREL, TUB). The simulations performed by DTU did not include a core growth model because the core expansion is estimated from a direct Biot–Savart calculation between vortex elements, carried out in an auxiliary mesh.

---

[1] The results from CENER are available from a later vortex age due to limitations in the data extraction from the simulation.

Additionally, the blade discretization used by each participant could affect the formation of the tip vortex, and consequently the initial core radius. The multiple approaches that can be used cause large differences in the prediction of the vortex core radius, from about 1 % to 7 % of the rotor radius. All the CFD simulations overestimate the core radius, even though the differences between the participants are limited to around 1 %. The deviation from the experimental data is probably due to an incorrect initial size of the core. For the ALM simulations (POLIMI, TUD, and UNIFI), the initial core size is not obtained by

solving the flow around the tip of the blade – as is the case for blade-resolved simulations– but is a function of the kernel size (Shives and Crawford, 2013; Melani et al., 2022) given a sufficient mesh refinement. Hence, possible improvements might be achieved by acting on this parameter.

    In terms of core growth: POLIMI and DTU predict an increase of 1.3 % and 0.2 % of the chord, respectively, while TUD and UNIFI show an almost constant core radius over the analyzed vortex ages. The small differences observed across the CFD

approaches are probably a consequence of the methodology used. Indeed, CFD simulations solve the flow within the core instead of imposing a growth model, as is the case for the FVW approaches. Additionally, the results suggest that the discretization used by the participants is satisfactory, as no excessive numerical diffusion is shown (Cormier et al., 2020).

    The tip vortex strength (Fig. 7, right) was evaluated following the methodology described in Sect. 6.1. The experimental results show that the tip vortex strength decreases with vortex age in the range analyzed. The FVW approaches show good agreement

in terms of vortex strength with the experimental data, showing a maximum difference of about 0.13 $m^2/s$ (10 %). Most of the participants underestimate the circulation, and only CENER, DTU, and NREL show larger strength of the tip vortex. Additionally, the reduction of tip vortex strength with vortex age predicted by the FVW simulations is comparable to the one observed in the experiment.

    Some differences are observed between the CFD results. The ALM simulations (POLIMI, UNIFI, and TUD) underpredict the

tip vortex strength from the experiment by a maximum of about 20 %. The RANS and LES approaches performed by UNIFI and TUD, respectively, showed almost identical results, both in terms of circulation values and dissipation rates, while POLIMI showed a 5 % increase in initial vortex strength and higher dissipation. In contrast, the hybrid approach by DTU showed the highest tip vortex strength of all, which might be caused by the smaller dissipation rate within the core.

    In conclusion, when analyzing the tip vortices in the near wake of the fixed-bottom case, it is observed that no clear advantage

is obtained when switching from a FVW approach to a CFD ALM simulation. The two methodologies provide similar results to the experimental tests in terms of streamwise position of the tip vortex and convection velocity if the effect of blockage is taken into account (see Sect. 4). In terms of core radius and strength, relevant differences are observed among the participants using FVW, which are probably due to the different setups employed by the participants; however, if the simulations are tuned appropriately, these approaches can correctly calculate these two metrics. Instead, further efforts are required to refine the CFD

ALM approaches, as all participants overpredicted the core radius and underestimated the strength of the tip vortex in comparison to the experimental result.

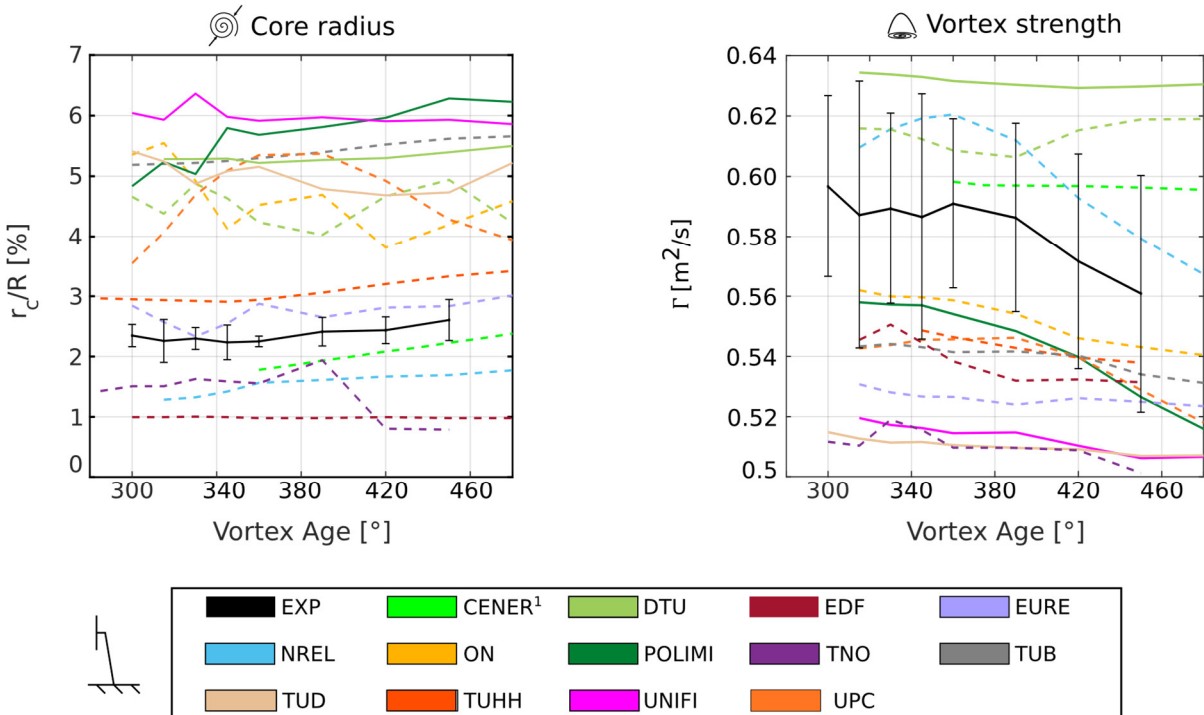

**Figure 7** Core radius (left) and vortex strength (right) as a function of vortex age for the fixed-bottom load case (LC1.1). Dashed lines represent the FVW results, and solid lines represent either the CFD or experimental results. Error bars represent the standard deviation of the experimental data.

### 7.2 Unsteady cases

In this section, the PIV results from both pitch and surge cases are presented. First, an in-depth analysis of LC2.5 is shown (Sect. 7.2.1). Then, the aggregated results from the rest of the LCs are analyzed to compare the effect of platform motion amplitude and frequency on the results. For surge cases, as discussed, no error bars could be added to the experimental data, as the velocity fields were acquired for a single cycle of platform motion. This represents one of the limitations of the current study, and further experimental tests are required in the future to validate the results shown in Sects. 7.2.1. and 7.2.2. The plots shown in Sect. 7.2.1 for LC2.5 are included as supplementary material to this work for the remaining LCs.

### 7.2.1 Load case 2.5

Figure 8 (a) shows the streamwise position of the tip vortex for LC2.5 at a vortex age of 427°. The results are shown over a cycle of platform motion. The zero-degrees position represents the start of the surge motion (i.e., turbine moving upstream at the fixed-bottom position $x = 0$). All the results shown are shifted to account for the convection time required by the tip vortices to reach the field of view (see Sect. 5.1) to allow a direct comparison with the platform motion. Hence, the negative angle values represent the tip vortices shed during the previous cycle of platform motion. In Fig. 8 (a) experimental data are represented as crosses due to the limited number of data points available, and the FVW and CFD results are plotted as dashed and solid lines, respectively. The amplitude and phase shift of the tip vortex streamwise position and strength is estimated from the only four available data points. For this reason, the reported values for these metrics could be affected by noise and aliasing and further experimental tests are required to confirm the preliminary results presented in this Section.

The tip vortex position shows an apparent motion due to the motion of the wind turbine relative to the PIV field of view, which remains fixed with respect to the ground. Hence, the position of the tip vortices in the inertial reference system should follow the platform motion if the position of the vortices relative to the wind turbine remains constant. The experimental results, however, show that the tip vortex is found further upstream compared to the steady case (indicated by the red diamonds in Fig. 8 (b)), suggesting that the surge motion influences the shedding and convection velocity of the vortices at early vortex ages, resulting in a slower initial convection of the tip vortices. Furthermore, the positive phase shift (Fig. 8 (c)) observed in the

experimental results indicates that the tip vortices are traveling faster downstream while the wind turbine is moving upstream compared to when the turbine is moving downstream, confirming what was already observed qualitatively by Fontanella et al. (2021) in previous work on the same experimental data set.

The FVW simulations show a quasi-steady behavior of the tip vortex streamwise position, which seems to oscillate around the steady-state position with a similar amplitude to the platform motion (Fig. 8 (b)). Interestingly, some of the FVW approaches show that the average position of the tip vortices is found slightly downstream compared to the steady-state case, even though the difference is limited compared to what is observed in the experimental results. According to Kleine et al. (2022), even small differences in tip vortex path can affect the stability of the wake and lead to faster collapse of the tip vortices; hence, modeling the position of the tip vortices is of paramount importance.

Concerning the phase shift, there is no agreement between the FVW simulations, as half of them predict a positive phase shift while the others predict a negative phase shift. The FVW simulations also show larger phase shifts compared to the experimental data. For a more thorough validation, more experimental data are required, as only a limited number of points per surge cycle are available. Additionally, the PIV data set includes only velocity fields from a single cycle of platform motion (see Sect. 3).

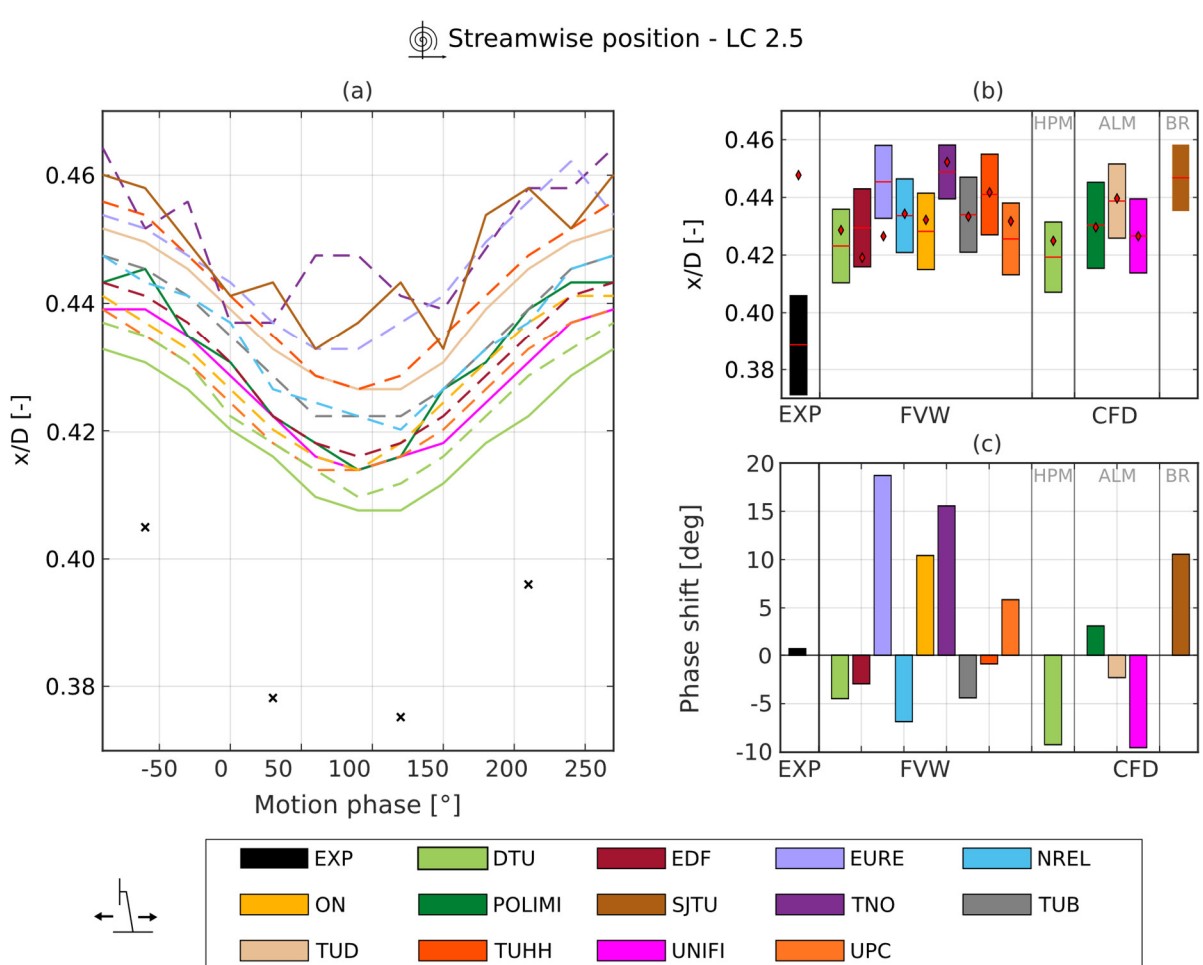

Figure 8 (a) Streamwise position of the tip vortex during a cycle of platform motion at a vortex age of 427°. Dashed lines represent the FVW results, solid lines represent the CFD results, and the black crosses represent the experimental data. (b) Oscillation amplitude of the streamwise position of the tip vortex. The box plot represents the amplitude, the red lines represent the average position of the tip vortex during the cycle of motion, and the red diamonds represent the average position in the fixed-bottom case. (c) Phase shift of tip vortex position with regard to platform motion.

The CFD results predict analogous results to the FVW methods: the tip vortex oscillates around the steady-state position with amplitude equal to the platform motion. No agreement is observed in terms of sign of the phase shift, but the difference from the quasi-steady value is up to approximately ±11°. The largest phase shift is shown by SJTU, which performed blade-resolved

simulations using an unsteady detached-eddy simulation turbulence model. In their results a higher-frequency oscillation is also shown, which is not present for the other CFD approaches.

Figure 9 shows the tip vortex strength during the surge cycle for LC2.5. The tip vortex strength follows a sinusoidal trend induced by the platform motion. During the surge cycle, the relative wind speed will be given by the sum of the freestream wind velocity and the velocity of the platform motion. Therefore, when the wind turbine is moving upstream, it operates at a higher relative wind speed and the blade loading increases, whereas the opposite happens when the turbine moves downstream, as the rotor is subjected to a lower relative wind speed. As blade loading increases, so does the pressure difference between

the suction and pressure side of the blade, and therefore so should the tip vortex strength. Hence, the changes in apparent wind speed result in a variation of the tip vortex strength, which follows the velocity profile of the platform. According to the linearized quasi-steady theory, the vortex strength should be phase-shifted by −90° compared to platform motion (see Sect. 2.1). Indeed, this trend is confirmed in Fig. 9 (a), where the tip vortex strength shows a sinusoidal behavior for most numerical approaches. The experimental results do not follow a sinusoidal trend in terms of vortex strength; however, this is probably

due to the limited number of points. Indeed, for LC2.1 a larger number of data points are available, and a sinusoidal trend can be identified. The amplitude and phase shift of the experimental results are shown for reference (Fig. 9 (b,c)), even though these values are probably not representative of the actual trend.

The tip vortex strength oscillations calculated from the experimental data do not seem to oscillate around the fixed-bottom value, and the average strength shows an increase of 18 % during the surge motion. A similar increase is also observed in the

640 rest of the available LCs. In contrast, all simulation approaches do not predict a similar increase in tip vortex strength, and the circulation oscillates around the steady-state value. Hence, all simulations underpredict the circulation of the tip vortices for the surge cases. As discussed in Sect. 7.1, good agreement is seen in fixed-bottom conditions between experiments and numerical models regarding tip-vortex strength. Therefore, this difference between the experimental and numerical results is probably not caused by blockage. This is confirmed by the CFD simulations, which include the wind tunnel walls (UNIFI,

POLIMI, TUD), as they do not show an increase of the circulation of the tip vortices. Furthermore, the cause does not seem to be connected to an increase in rotor loading, since the rotor thrust measurements did not show a similar rise; instead the rotor thrust oscillated approximately around its steady-state value (Bergua et al., 2023). The variations in relative velocity might affect the roll-up of the tip vortex and cause a further interaction between the tip vortices and the trailing vorticity, resulting in a higher strength of the tip vortices. Nevertheless, given the large increase in tip vortex strength and the significant difference

observed with regard to the simulations, it is possible that the experimental data are not reliable in this case – further testing should be carried out to confirm these results.

The FVW simulations predict larger tip vortex strength oscillations compared to CFD, but the oscillation amplitude can vary significantly (by up to 75 %) across the participants. Similarly, a difference of up to 28 % is observed across the CFD approaches.

In terms of phase shift, the FVW approaches show good agreement, as all the participants predict limited differences ($\Delta\varphi <$ 20°) from the quasi-steady value of −90°. The majority of FVW simulations show a larger phase shift compared to the linearized quasi-steady theory, and only EDF and TNO predict a phase shift smaller than 90°.

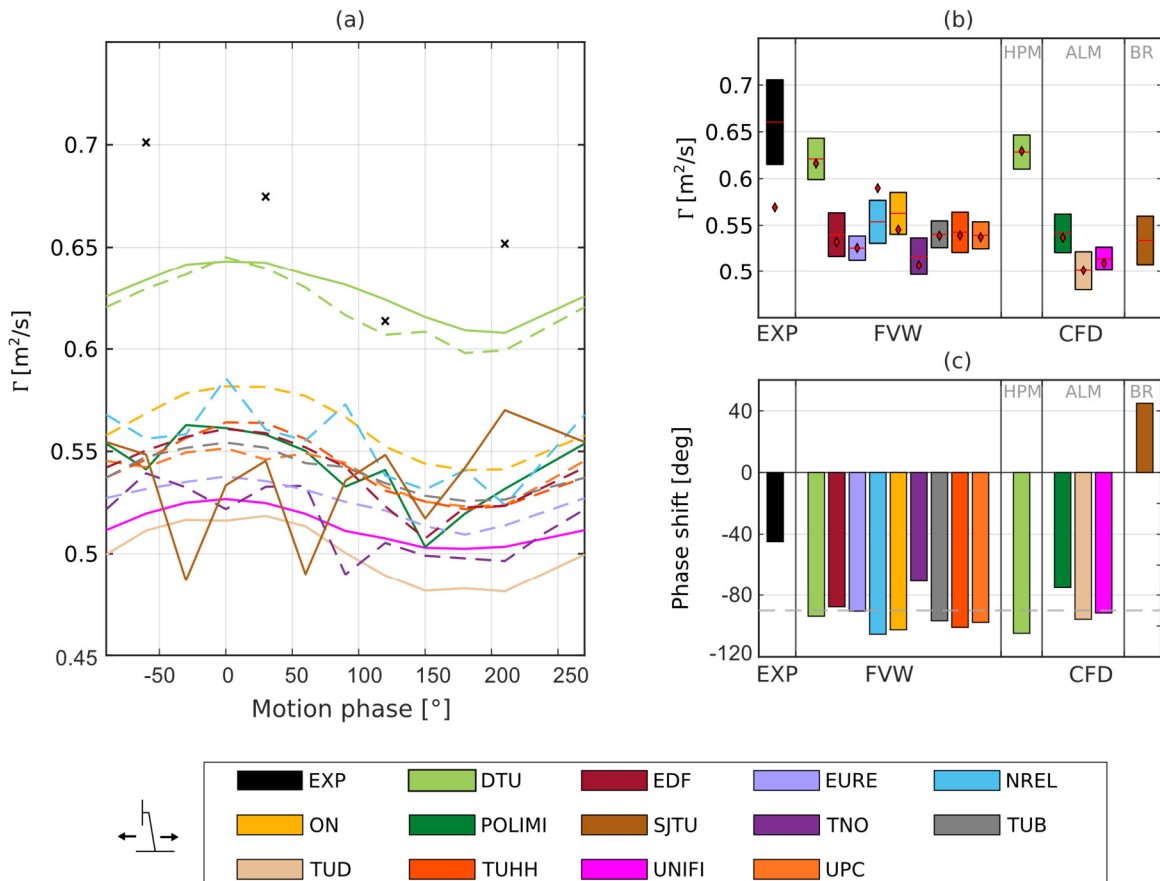

**Figure 9 (a) Tip vortex strength during platform motion at a vortex age of 427°. Dashed lines represent the FVW results, solid lines represent the CFD results, and the black crosses represent the experimental data. (b) Amplitude of strength oscillations at the frequency of platform motion. The red lines represent the average tip vortex strength during a cycle of motion, and the red diamonds represent the corresponding value from the fixed-bottom case. (c) Phase shift of the tip vortex strength with regard to platform motion.**

Similarly, most of the CFD results either predict a small advance or delay from 90°, with similar scattering compared to the FVW approaches. The exception is represented by SJTU, which shows a phase shift of about 45°. Additionally, the tip vortex strength trend shows the superposition of higher-frequency oscillations, which are not present in the other CFD results.

Finally, the surge results can be compared to the fixed-bottom results in terms of core radius. Figure 10 (a) shows the evolution of the core radius during the surge motion at a vortex age of 427°. For each participant, the average core radius is calculated and shown by the bars in Fig. 10 (b). In the same figure, the whiskers represent the minimum and maximum values of the core radius during the cycle of motion, and the red diamonds indicate the fixed-bottom value. From the experimental results, the surge motion appears to have no impact on the core radius, as almost no oscillation is observed, and the average value is equal to the fixed-bottom value. Among the FVW simulations, most of the participants do not present a variation of the average core radius in comparison to the fixed-bottom value. However, this is not the case for the results obtained by NREL, ON, and TNO, which show an increase of the average core radius. The same trend is not shown in the CFD results (DTU, POLIMI, UNIFI, and TUD), as only minimal differences from the fixed-bottom value are observed for these higher-fidelity methodologies. For the blade-resolved simulations performed by SJTU, no data are available for the fixed-bottom case, and the same comparison cannot be carried out.

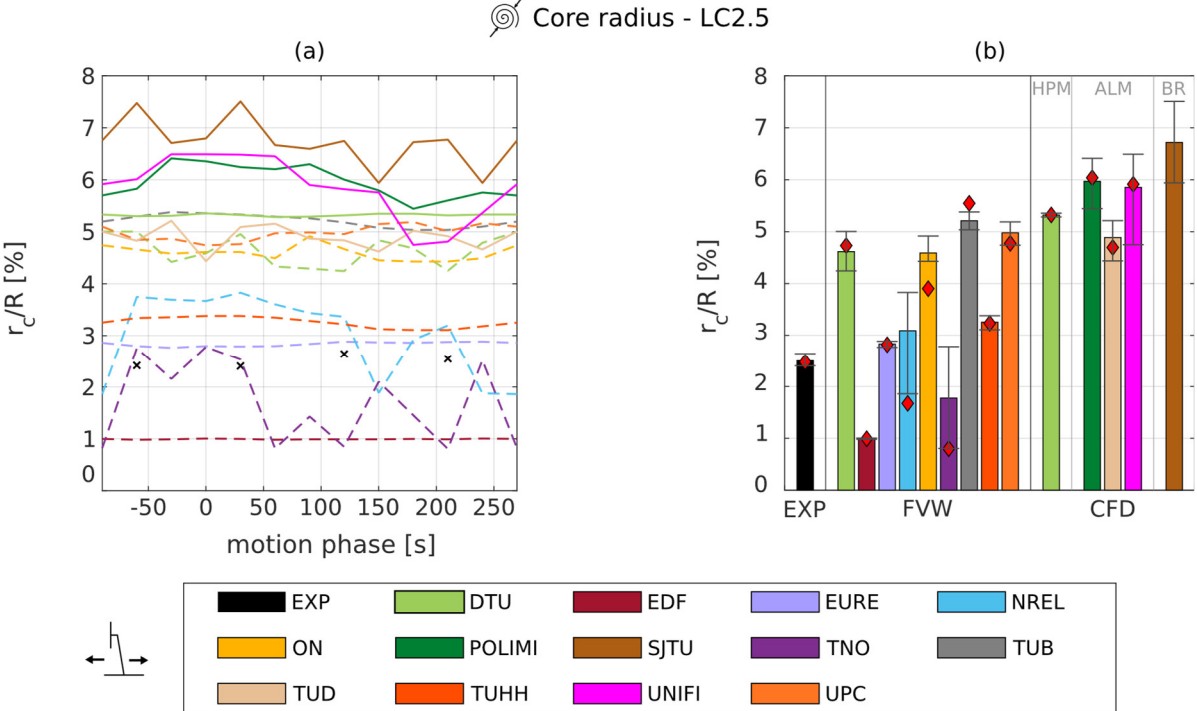

**Figure 10 (a) Tip vortex core radius during platform motion. Dashed lines represent the FVW results, solid lines represent the CFD results, and the black crosses represent the experimental data. (b) Average core radius during surge motion. The whiskers represent the minimum and maximum values of the core radius during the cycle of motion, and the red diamonds indicate the fixed-bottom value.**

Additionally, some of the FVW and CFD participants (EURE, TUHH, UPC, POLIMI, SJTU, UNIFI) predict small oscillations in core radius at the frequency of platform motion; however, their amplitude is limited. Such oscillations are present even though these approaches do not solve the formation of the tip vortex, as the FVW methodologies impose the initial core size, and for the ALM simulations this parameter is a function of the kernel size.

The variations in core radius observed in the simulations could indicate that the surge motion of the platform alters the roll-up of the tip vortex, as differences in the blade tip loading could alter the mixing between the tip vortex and the trailing vorticity.

### 7.2.2. LC summary

The analysis carried out in the previous section for LC2.5 is replicated for the rest of the surge and pitch cases. In this way, the effect of the direction (surge or pitch) and frequency of motion on the results can be analyzed. The results are shown in terms of streamwise position and strength of the tip vortices as these are the metrics that are most affected by the imposed platform motion. The results for the LCs that are presented here only in an aggregated form, are available as supplementary material to this work.

In Fig. 11 (a), the amplitude of the streamwise position of the tip vortex is shown as a function of the amplitude of platform motion for the surge cases. The simulation results agree with the experiment for LC2.5 up to about 0.005D (0.01m); however, the experimental results diverge from the expected quasi-steady value for LC2.1. Indeed, according to quasi-steady theory, the position of the tip vortices relative to the turbine should remain constant. Hence, in the inertial reference frame, the tip vortices should oscillate around their fixed-bottom position with the same amplitude and phase of the sinusoid of platform motion. The analysis of LC2.1 showed that the streamwise position of the tip vortex is not periodic and does not follow a sinusoid at the frequency of platform motion, affecting the amplitude and phase shift. Since the experimental data were captured only for a single cycle of platform motion, this result might be an outlier, and further data are required to confirm the insurgence of any unexpected behavior of the tip vortices.

The amplitude of the tip vortex streamwise oscillations increases linearly with the platform motion amplitude for most of the FVW and CFD participants, meaning that for the analyzed LCs, the tip vortex follows the motion of the turbine with only

minor differences. Conversely, differences are observed in terms of phase shift compared to platform motion. At the lowest frequency (LC2.1) all simulation approaches yield a small phase shift ($\Delta\varphi \approx 0$). However, the scattering of the simulation results increases with the reduced frequency, and differences increase across the participants. As differences increase, the numerical models are farther from the linearized quasi-steady solution. The streamwise position of the tip vortex depends on the freestream velocity and the flow velocity in the wake: the higher the velocity reduction, the slower the tip vortex is convected downstream. Therefore, differences in induced velocity on the rotor, which affect the convection velocity and formation process of the tip vortex, could explain the scattering that is observed when the surge frequency is increased. This result confirms the threshold proposed by Fontanella et al. (2021) for rotor forces, as the aerodynamic response of the rotor diverges from the linearized quasi-steady theory when the reduced frequency is larger than 0.5.

Additionally, the participants that used unsteady airfoil aerodynamics (DTU, NREL, UPC, TNO, TUB) tend to predict slightly larger phase shifts compared to those that used the steady polars (ON, TUHH). However, this trend is not reflected for all the participants, as EURE shows similar phase shifts to the participants using steady polars, and EDF predicts only minimal phase shifts at the highest frequency. For this reason, the FVW results suggest that the discrepancies observed in the tip vortex position from the linearized quasi-steady theory are not driven by airfoil-level unsteady effects (as described in Sect. 2.2). Indeed, the airfoil reduced frequency overcomes the threshold (see Eq. (14)) proposed by Sebastian and Lackner (2013), only for the inner parts of the blade (approximately $\frac{r}{R} < 0.4$) and not in the tip region.

Among the CFD results, the LES simulations by TUD show a positive phase shift of about 20° for LC2.7, in agreement with some of the FVW approaches. Instead, the phase shift is limited to less than 7° for the RANS simulations by UNIFI and the hybrid approach by TUD, which both use an ALM approach with steady polars. In contrast, POLIMI shows a sharp increase in phase shift for LC2.7, despite performing the simulations with an analogous methodology to UNIFI. The blade-resolved approach used by SJTU seems to confirm the trend shown by some of the FVW methods (TNO, EURE ON) for LC2.5; however, for LC2.7 the predicted phase shift does not show a significant increase. Hence, no clear trend can be observed from the CFD ALM simulations, and the results show relevant differences with the blade-resolved data obtained by SJTU. This might indicate that the ALM methodology is not able to capture the behavior of the tip vortices correctly when the reduced frequency increases.

The experimental results show a phase shift of less than 1° for LC2.5, but no data are available for LC2.7 to carry out a direct comparison with the simulations.

Figure 11 (c, d) shows the amplitude and phase shift of the streamwise position of the tip vortex for the pitch cases. The results are plotted as a function of the streamwise amplitude of the blade tip motion $A_x$. Compared to the surge case, further differences are observed from the expected quasi-steady value for the amplitude of the streamwise position of the tip vortex. While the simulation approaches show good agreement with the quasi-steady theory for LC3.5 and LC3.7, the results diverge from the expected linear trend for LC3.1, in contrast to the surge cases.

In terms of phase shift, no major differences are observed compared to the surge cases. The majority of the FVW participants predict small phase shifts for LCs 3.1 and 3.5 and then the scattering of the results increases for LC3.7. The phase shift can be either positive or negative, depending on the participant. All participants, except TNO and EURE, predict similar phase shifts for the equivalent pitch and surge LCs. Among the CFD participants, only minor differences are observed between the surge and pitch cases.

Analogously to the results shown for the tip vortex streamwise amplitude, Fig. 12 shows the oscillations of the tip vortex strength as a function of the frequency of platform motion. This visualization is chosen because the amplitude of the strength oscillations is a function of the frequency of platform motion (see Sect. 2.1). The majority of FVW and CFD codes predict an almost linear increase of the tip vortex strength amplitude with the frequency of motion, in agreement with the linearized quasi-steady theory (Sect. 2.1). Comparing the experimental and simulation results, the former shows a larger increase in tip vortex strength amplitude with reduced frequency.

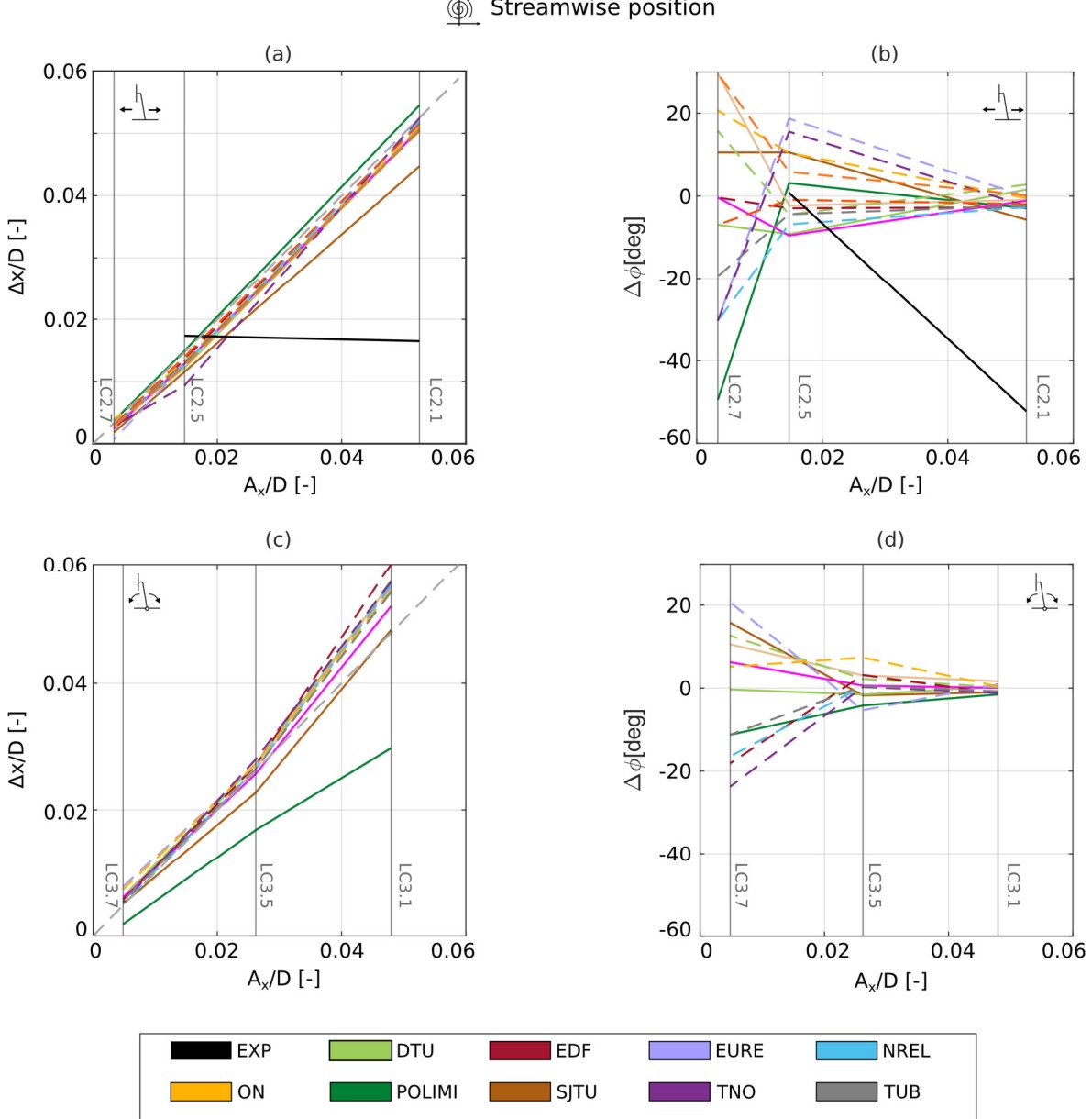

**Figure 11 Tip vortex streamwise position amplitude and phase shift for surge (a, b) and pitch (c, d) cases. Dashed lines represent the FVW results and solid lines represent either the CFD or experimental results.**

Additionally, some participants (POLIMI, SJTU, TNO, UPC) show a clear divergence from the linear trend at the highest reduced frequency (LC2.7). However, the aerodynamic response varies among the participants. SJTU shows a sharp decrease of the normalized strength oscillation amplitude, while POLIMI and TNO show an almost constant trend at the highest frequencies.

Among the FVW and CFD approaches showing a linear increase of the strength amplitude, the predictions of the slope of the curve vary significantly. The slope is a function of the induction factor and of the partial derivative of the circulation with respect to the relative velocity (Eq. (8)) and should represent a characteristic of the tested rotor geometry. No clear trend can be distinguished when comparing the FVW approaches using steady polars or unsteady airfoil aerodynamics, similar to what was already observed in terms of streamwise position of the tip vortices, suggesting that the effect of airfoil-level unsteadiness on the circulation of the tip vortices is limited.

The CFD approaches provide comparable results to the FVW methods; the blade-resolved simulations by SJTU, which represent the highest-fidelity approach, predict the largest increases in normalized strength oscillation for LC2.5, closer to the experimental results. This might indicate that FVW and ALM approaches could underestimate the effect of platform motion

on the strength of the tip vortices. However, further experimental tests at high reduced frequency are required to validate these results.

In terms of phase shift, the linearized quasi-steady theory predicts a delay of the tip vortex strength oscillations of 90° from platform motion (see Sect. 2.1). The experimental results suggest a phase shift of about 50° for LC2.1 and a further decrease is observed for LC2.5. Hence, the linearized quasi-steady theory might not be capable of predicting the phase shift of the strength of the tip vortices. Nevertheless, given the limited number of available experimental points, and given that the experimental velocity fields were captured only for a single cycle of surge motion, further tests at high reduced frequency are required to validate these results.

For LCs 2.1 and 2.5, the majority of the FVW approaches show differences of up to 20° from the expected linearized quasi-steady value, and only minimal scattering is observed across the participants. For LC2.7, a sharp increase in tip vortex strength phase shift can be noticed for most participants. The phase shift reaches values of up to −130°, which means that the tip vortex strength follows an almost opposite trend to the expected one (i.e., the tip vortex strength increases when the turbine is moving downstream).

The CFD results show further differences across the participants. The hybrid approach by DTU and the ALM simulations by TUD show a −90° phase shift for LC2.1 and only a slight increase with the reduced frequency to about −100°. In contrast, the ALM simulations performed by UNIFI show the tip vortex strength in advance with respect to platform motion ($\Delta\varphi > -90°$), both at low and high reduced frequencies. At high frequencies, the phase shift shows an opposite trend compared to the FVW approaches. At low frequencies (and high amplitudes), the differences from the linearized quasi-steady theory should be minimal. The ALM results from POLIMI also show the phase shift in advance at low reduced frequency, as is the case for the experimental data; however, the phase shift increases up to -150° at high reduced frequency.

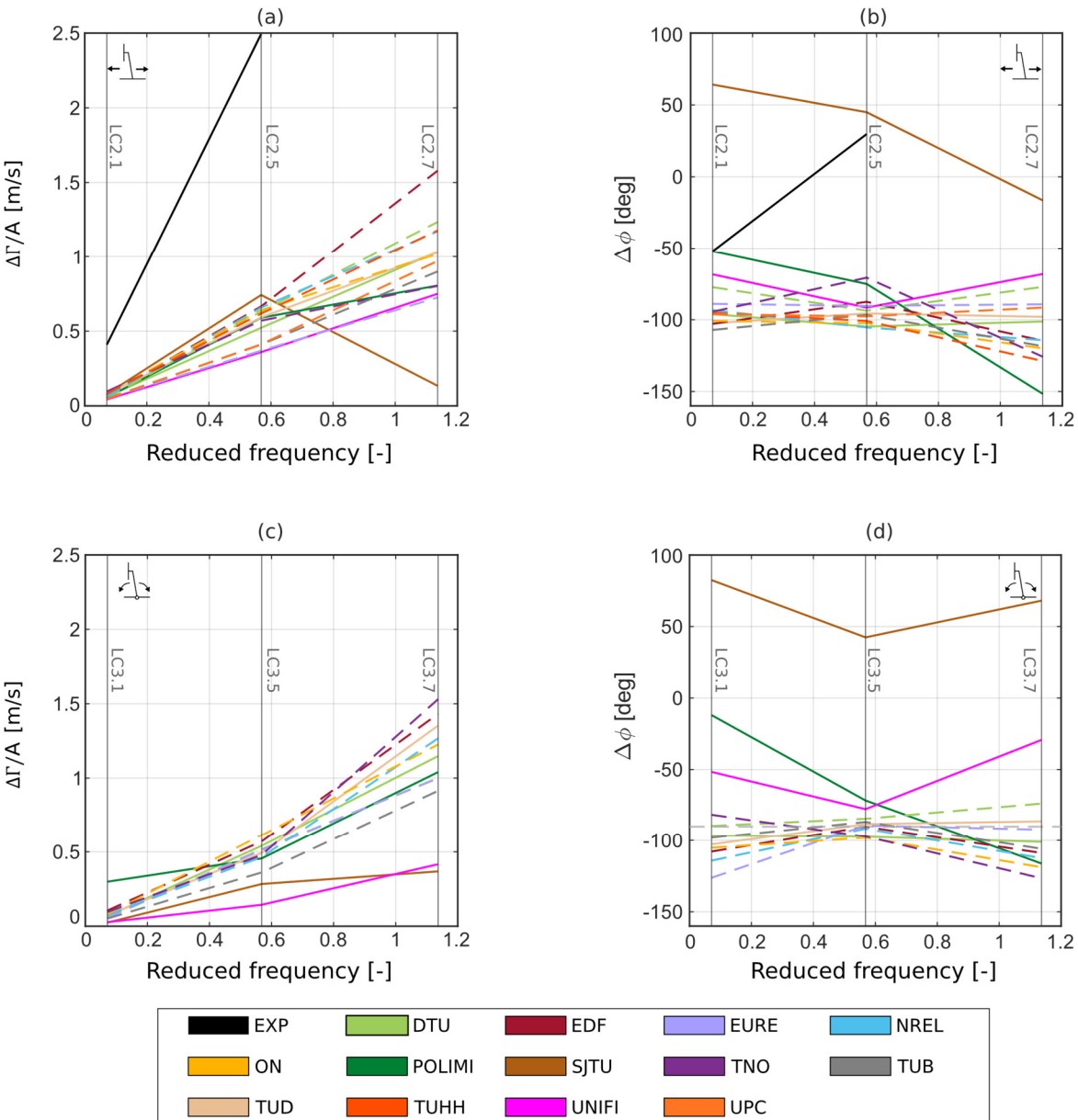

**Figure 12 Amplitude and phase shift of tip vortex strength for surge (a, b) and pitch (c, d) cases. Dashed lines represent the FVW results and solid lines represent either the CFD or experimental results.**

Discrepancies from quasi-steady theory are also observed in the blade-resolved results obtained by SJTU. In this case, the phase shift is positive for all LCs varying from about 80° to 70° for LCs 2.1 and 2.7, respectively. At the moment, no clear reason can be identified to explain this difference with the rest of the participants. Further experimental tests and simulations should be carried out to validate this result.

Figure 12 (c, d) shows the tip vortex strength amplitude and phase shift for the pitch cases. In contrast to what is observed for the surge cases, most of the simulations do not predict a linear increase of the vortex strength with the amplitude, suggesting that the pitch motion might affect the behavior of the tip vortices more compared to the surge motion, confirming what was already observed from the analysis of the streamwise position of the tip vortices. Otherwise, the simulations might not be able to reproduce the behavior of the tip vortices under a pitching motion of the platform. For this reason, experimental tests should be carried out to validate these results.

Additionally, the phase shift results present some differences compared to the surge cases, especially at the lowest frequency of motion. The FVW approaches, which showed almost no phase shift from the quasi-steady value for the corresponding surge motion case, indicate an increase in the phase shift, even if not for all the participants. Similarly, the ALM RANS results by

POLIMI and UNIFI show larger differences from the −90° phase shift compared to the surge case. This result suggests that for a pitch motion of the platform the behavior of the tip vortices might diverge from the linearized quasi-steady theory, even at low reduced frequencies. A possible cause might be that the pitch motion does not only affect the relative wind speed seen by the rotor but also the wake shape. At the lowest frequency of motion, the rotor reaches the largest tilt angles, which affect the geometry of the tip vortex helix (e.g., the spacing between tip vortices), as already observed for fixed-bottom turbines by Wang et al. (2020). The discrepancy observed at large amplitudes of motion could also be caused by the asymmetry in the pitching motion of the rotor, due to the tilted tower configuration (see Fig. 1). Indeed, the trajectory of the blade tip is not symmetric around the fixed-bottom position but varies for positive and negative pitch angles. At large amplitudes of motion, the asymmetry becomes more pronounced and could contribute to the observed differences in terms of tip vortex position and strength. Since no PIV data are available for the pitch cases, future experimental data could aid the understanding and validation of these results.

After analyzing tip vortex characteristics as a function of platform motion amplitude and frequency, it is also interesting to investigate whether a relationship between tip vortex characteristics and rotor loads can be found. Platform motion causes tip vortex strength variations as well as rotor thrust oscillations (Bergua et al., 2023). Since the tip vortex strength depends on the loading of the rotor, an increase in rotor thrust oscillations should be reflected in an increase of the tip vortex strength amplitude. As the thrust and vortex strength amplitudes increase linearly with the reduced frequency, the two metrics show a linear relationship (Fig. 13 (a)). However, a collapse of the curves onto a single line is not observed, suggesting that the differences observed in terms of tip vortex strength may not be related to differences in predicted rotor loading.

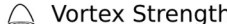

Figure 13 shows four plots (a), (b), (c), (d) with the following axis labels:

- (a): y-axis $\Delta\Gamma/A$ [m/s] (0 to 2.5), x-axis $\Delta T/A$ [N/m] (0 to 250). Vertical markers: LC2.1, LC2.5, LC2.7
- (b): y-axis $\Delta\varphi(\Gamma)\text{-}\Delta\varphi(T)$ [deg] (−40 to 120), x-axis Reduced frequency [-] (0 to 1.2). Vertical markers: LC2.1, LC2.5, LC2.7
- (c): y-axis $\Delta\Gamma/A$ [m/s] (0 to 2.5), x-axis $\Delta T/A$ [N/m] (0 to 250). Vertical markers: LC3.1, LC3.5, LC3.7
- (d): y-axis $\Delta\varphi(\Gamma)\text{-}\Delta\varphi(T)$ [deg] (−40 to 120), x-axis Reduced frequency [-] (0 to 1.2). Vertical markers: LC3.1, LC3.5, LC3.7

Legend: EXP, DTU, EDF, EURE, NREL, ON, POLIMI, SJTU, TNO, TUB, TUD, TUHH, UNIFI, UPC

**Figure 13** Amplitude and phase shift of tip vortex strength with regard to rotor thrust in surge (a, b) and pitch (c,d). Dashed lines represent the FVW results and solid lines represent either the CFD or experimental results.

Additionally, it is useful to compare the phase of the thrust and strength oscillations (Fig. 13 (b)). Since both oscillations should follow the velocity profile of the platform (i.e., they should present a −90° phase shift compared to platform motion), they should be in phase. Indeed, all simulation results oscillate around the expected value; however, some scattering is present, as was already observed in Fig. 12. The main reason is that the thrust oscillations are generally in phase with the platform velocity and only small phase shifts are observed, as was shown by Bergua et al. (2023). In terms of rotor thrust, the discrepancies from the quasi-steady value increase with the reduced frequency similarly to what is observed in terms of tip vortex strength, but the scattering is generally limited to less than 10°. The discrepancies observed in terms of amplitude and phase shift could be caused by differences in the aerodynamic response along the spanwise direction of the rotor. Indeed, the amplitude and phase shift of the rotor thrust are a function of the aerodynamic response along the whole blade, while the amplitude and phase shift of the tip vortex strength are only a function of the aerodynamic response at the tip. Otherwise, these discrepancies may be caused by differences in the aerodynamic modeling of the wake, as performed by the tested numerical methods.

In conclusion, the results show how for a surge motion of the platform, most of FVW and CFD simulations provide similar results at low reduced frequencies. However, when the reduced frequency rises over 0.5, the scattering of the simulation increases, and significant differences are observed across the participants. Additionally, for the pitch cases, the differences

across the numerical approaches are also found at low reduced frequency. These results suggest that further tuning of the numerical approaches is required to improve the reliability of the simulations under surge and pitching motions of the platform.

## 8 HWA results

In this section, the results from the HWAs are presented. The results for LC2.5 are shown first, followed by a description of
the results for all surge and pitch cases. The plots shown in Sect. 8.1 for LC2.5 are included as supplementary material to this work for the remaining LCs. The fixed-bottom results were described by Bergua et al. (2023).

### 8.1 LC2.5

The surge and pitch motions cause a streamwise velocity perturbation in the wake induced by the changes in rotor thrust (Kleine et al., 2022; Fontanella et al., 2022b; Ramos-García et al., 2022). For example, Fig. 14 shows the streamwise velocity
for the along-wind HWA at $x = 5.48$ m and $y = 0.9$ m. By applying the Fourier transform, the amplitude of the streamwise velocity oscillation at the frequency of platform motion can be extracted. The surge motion does not seem to affect the mean velocity values, as both experimental and simulation results oscillate close to the steady-state value. The only outlier is represented by the FVW simulations by EURE, which show a larger velocity value for the unsteady case compared to the fixed-bottom one, which indicates a faster wake recovery (see Fig. 18).
Both FVW and CFD predict larger oscillations amplitude compared to the experimental results, indicating that the effect of the platform motion on the wake is overestimated for both simulation methodologies.

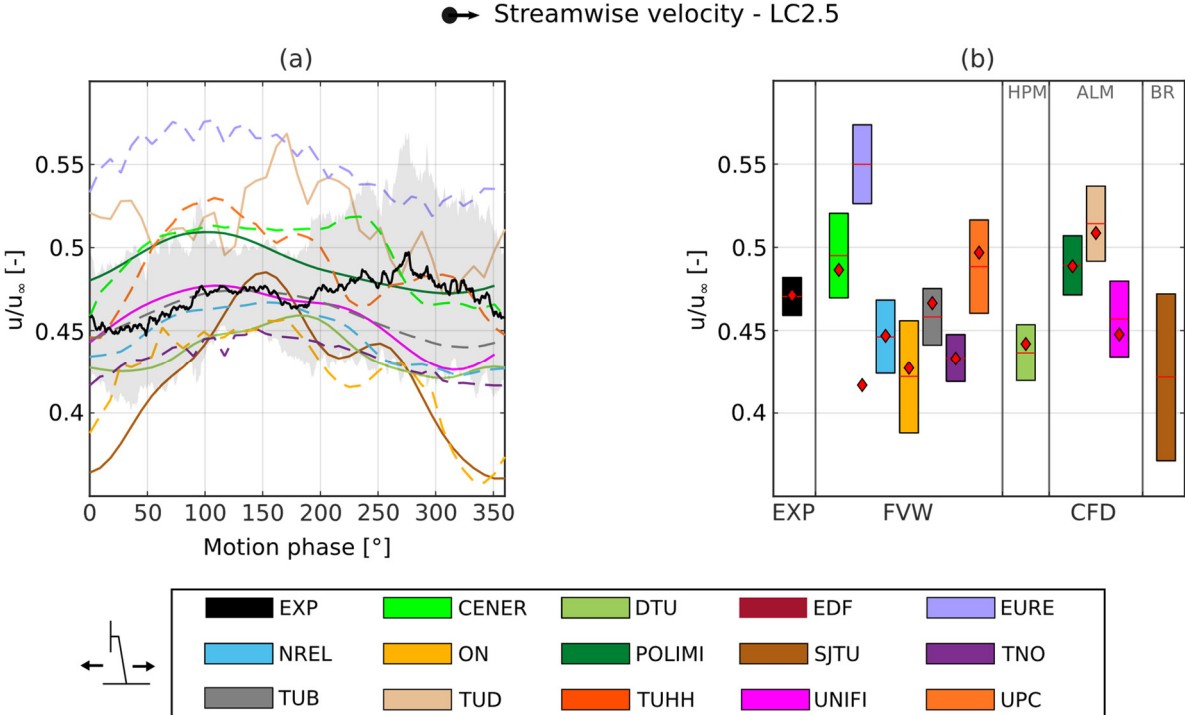

**Figure 14 (a) Streamwise velocity oscillations during platform motion from a single HWA probe at x = 5.48 m and y = 0.9 m for LC2.5. Dashed lines represent the FVW results and solid lines represent either the CFD or experimental results. The shaded area**
**represents the standard deviation of the experimental data (b) Amplitude of the streamwise velocity oscillations calculated at the frequency of platform motion. The red lines represent the average values during the cycle of motion, and the red diamonds represent the fixed-bottom case.**

The effect of platform motion on the wake can also be observed in terms of wake deficit (see Sect. 6.2), as the variation in
rotor loading affects the amount of energy extracted from the wind and hence the velocity deficit in the wake. Figure 15 shows the wake deficit oscillations during a cycle of surge motion for LC2.5. Indeed, the motion of the rotor does not only induce a

velocity oscillation in specific points of the wake, but the effect is also visible when considering the wake deficit, which includes the contributions from all the crosswind probes. Both the experimental and simulation results oscillate around the fixed-bottom value, represented by the red diamonds. Therefore, the differences observed between the average wake deficit values from experimental and simulation results are not induced by inaccurate wake modeling during platform motion. As already observed by Bergua et al. (2023) the FVW results probably differ because of the different wake lengths imposed during the simulations (Table 3). Indeed, if the wake recovery happens closer to the rotor, the recorded wake deficit will be lower, and vice versa. As was the case for the streamwise velocity oscillations, all simulation approaches predict larger amplitudes of the oscillations of the wake deficit compared to the experimental results. In terms of phase shift, most simulation results show the minimum value of the wake deficit in advance in comparison to the experimental results. The phase of the wake deficit is a function of the propagation speed of the velocity oscillations, which in turn is related to the average steady-state wake deficit (represented by the red diamonds in Fig. 15 (b)). The significant differences observed among the participants in terms of steady-state wake deficit could explain the discrepancies observed in terms of phase shift.

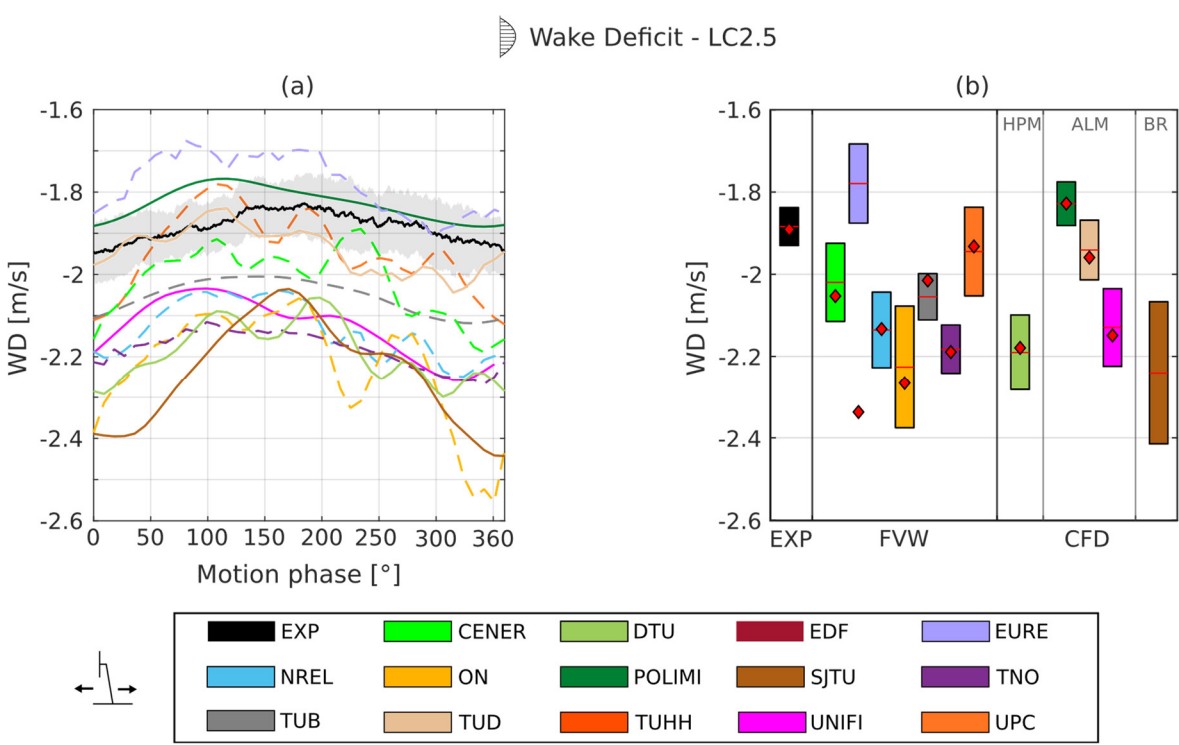

**Figure 15 (a) Wake deficit oscillations during platform motion for LC2.5. Dashed lines represent the FVW results and solid lines represent either the CFD or experimental results. The shaded area represents the standard deviation of the experimental data (b) Amplitude of the wake deficit oscillations at the frequency of platform motion. The red lines represent the average wake deficit during the cycle of motion, and the red diamonds represent the fixed-bottom value.**

## 8.2 LCs summary

The analysis of the wake deficit oscillations carried out in the previous section was repeated for all the LCs, and the results are shown in Fig. 16. According to the linearized quasi-steady theory (see Sect. 2.1), the amplitude of the wake deficit oscillations should increase linearly as a function of the frequency of platform motion. Indeed, the experimental wake deficit measured on the cross-plane line follows the quasi-steady linear trend, but some differences are observed in the simulation results. In fact, the majority of the FVW and CFD simulations predict a sharp increase in the wake deficit oscillations (due to an increase of the streamwise velocity oscillations) for LC2.7. TNO and POLIMI show the closest results to the experimental data for LC2.1 and 2.5, but a similar sharp increment is observed for LC2.7.

This difference between the experimental and simulation results is not justified by a similar increase in rotor thrust for LC2.7 (see Fig. 16 (b)), suggesting that the oscillations in the wake deficit are caused by the amplification of wake instabilities. As

noted by Kleine et al. (2022), the platform motion may excite vortex instabilities and cause the interaction of multiple vortices downstream, which result in large velocity oscillations. This effect should be more pronounced when the frequency of platform motion is half of the rotational frequency of the rotor (as is the case for LC2.7 and 3.7) (Kleine et al., 2022). However, the experimental data do not show a similar increase in the velocity oscillations for the tested LCs, and the wake deficit oscillations follow the linearize quasi-steady theory. Hence, the simulations might overpredict the growth of such instabilities. These differences could be related to unsteadiness in wind tunnel inflow, which are not captured by simulations and might dampen or modify the effect of these instabilities in the experimental results. However, this aspect was not investigated in the current study. Additionally, further experimental tests are required to confirm the available experimental data as, due to the limited number of data points available, error bars for the amplitude of the wake deficit of the experimental data could not be included. Comparing the surge and pitch results (Fig. 16 (c, d)), all participants predict similar wake deficit amplitudes between the two types of platform motion. Additionally, the increase in wake deficit amplitudes shown at the highest frequency of surge motion is also present for the pitch motion, suggesting that the far wake response is similar for the two cases.

The wake response to the surge platform motion can be observed in Fig. 17, where the amplitude of the streamwise velocity oscillations calculated from each of the along-wind HWA is shown as a function of rotor distance. The plots are presented only for the surge cases for brevity, but the pitch load cases provide analogous results. By comparing the velocity oscillations for increasing reduced frequency, the amplification of wake instabilities can be clearly observed. For the experimental results, the amplitude is almost constant and does not show a significant rise with increasing distance from the rotor for all LCs. With increasing reduced frequency, from LC2.1 to LC2.5 and LC2.7, the amplitude of the velocity oscillations grows by about two times. Among the simulation results, most participants predict similar amplitudes for LC2.1 (Fig. 17 (a)). However, for LC2.5 and especially for LC2.7, only CENER, TNO, and POLIMI, show comparable amplitudes to the experiments. The rest of the participants predict a significant increase (up to one order of magnitude) of the oscillation amplitude, independently of the methodology used. Additionally, large differences are observed across the participants, as the entity of the velocity oscillation varies from a maximum of 0.33 m/s to a minimum of 0.06 m/s.

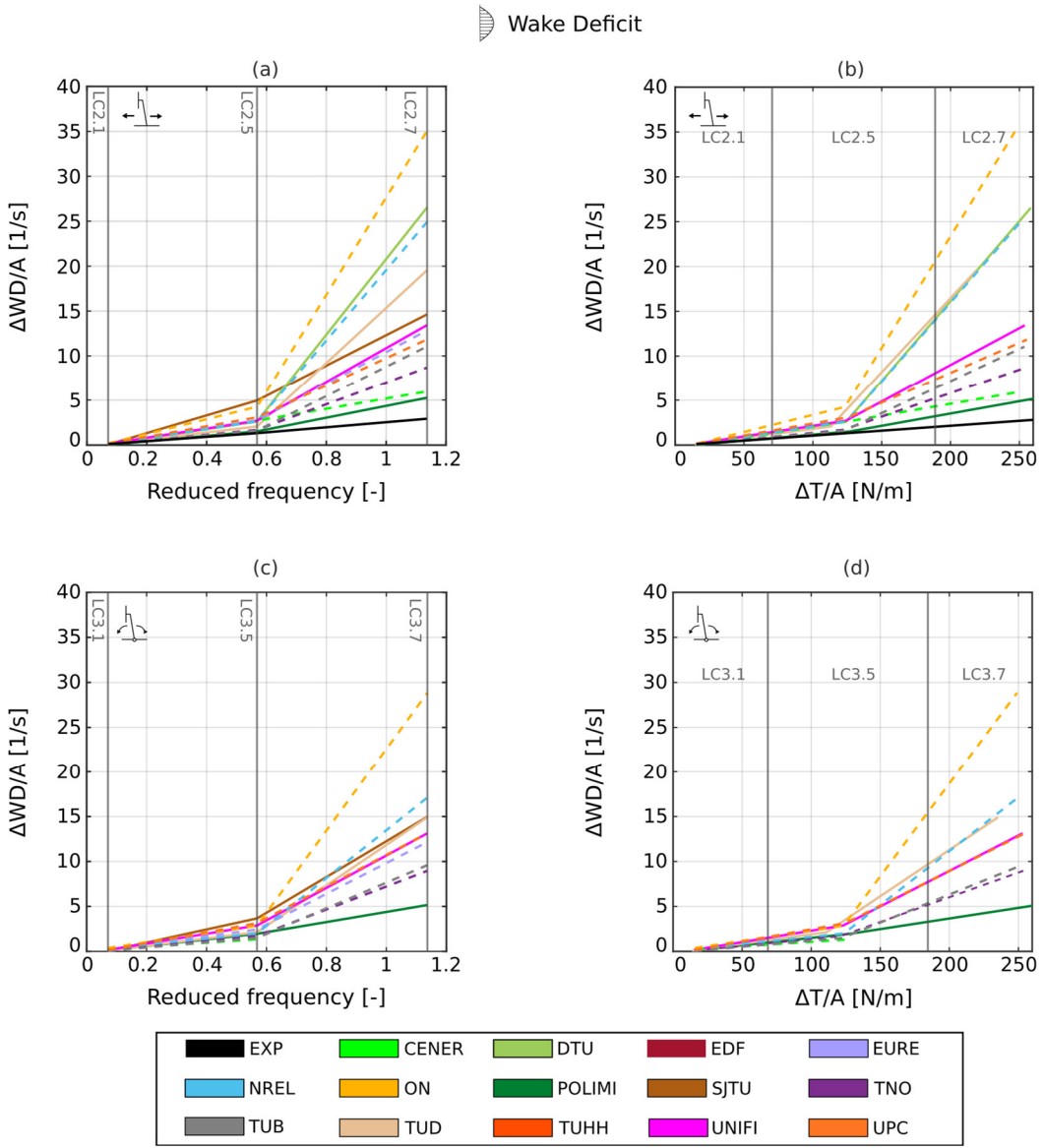

**Figure 16 Amplitude of normalized wake deficit oscillations as a function of reduced frequency and rotor thrust for surge (a, b) and pitch cases (c, d). Dashed lines represent the FVW results and solid lines represent either the CFD or experimental results.**

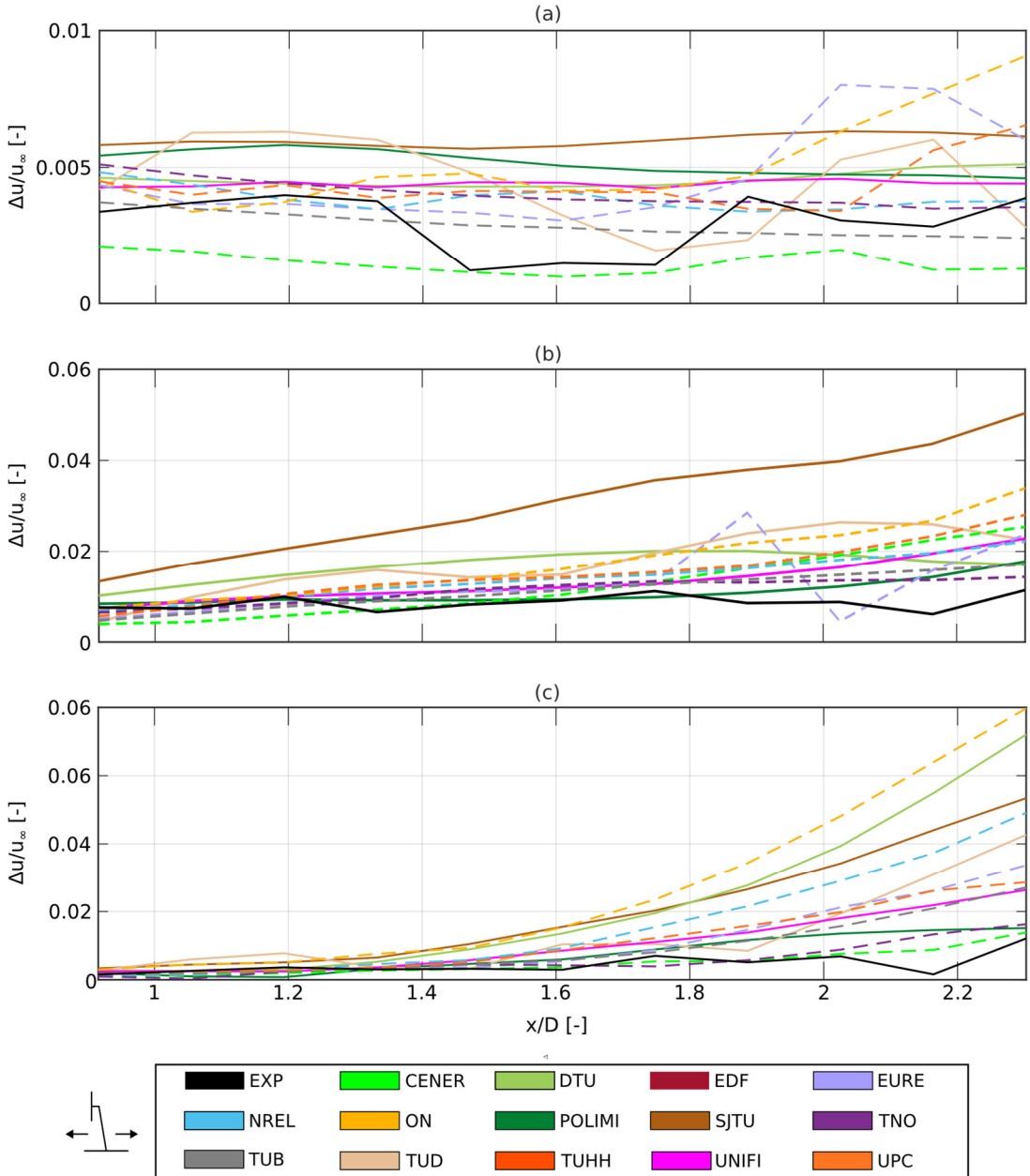

**Figure 17 Amplitude of streamwise velocity oscillations at the frequency of platform motion calculated from each of the along-wind probes as a function of rotor distance for the analyzed surge cases. (a) LC2.1: $A = 0.125$ m, $f_r = 0.071$, (b) LC2.5: $A = 0.035$ m, $f_r = 0.568$, (c) LC2.7: $A = 0.008$ m, $f_r = 1.137$. Dashed lines represent the FVW results and solid lines represent either the CFD or experimental results.**

Finally, Fig. 18 shows the average streamwise velocity value as a function of rotor distance for both surge and pitch cases. The experimental surge cases show that the wake recovery happens at about $x \approx 3.5$ m from the rotor. The location of the velocity minimum and the velocity trend are not affected by the surge motion, independent of the reduced frequency. Indeed, almost no difference is observed with the fixed-bottom case, which was presented by Bergua et al. (2023). Only the FVW simulations by EURE and UPC show the wake recovery at a similar distance from the rotor, whereas for the rest of the participants the velocity minimum is not observed before the last along-wind HWA. The differences observed across the FVW participants are most likely due to the imposed wake length in the simulation, which will affect the wake recovery. Despite this, all simulations show that the surge motion only has minimal impact on the wake recovery, for all the reduced frequencies analyzed. However, this is not the case for the pitch cases, where the average velocity decreases faster and recovers slower at higher reduced frequency for most of the participants. The trend cannot be validated through experimental data, but it is

consistent across the various simulation methodologies. As it was already observed during the analysis of tip vortices, the pitch motion of the platform seems to affect the behavior of the wake more profoundly than if affects surge cases.

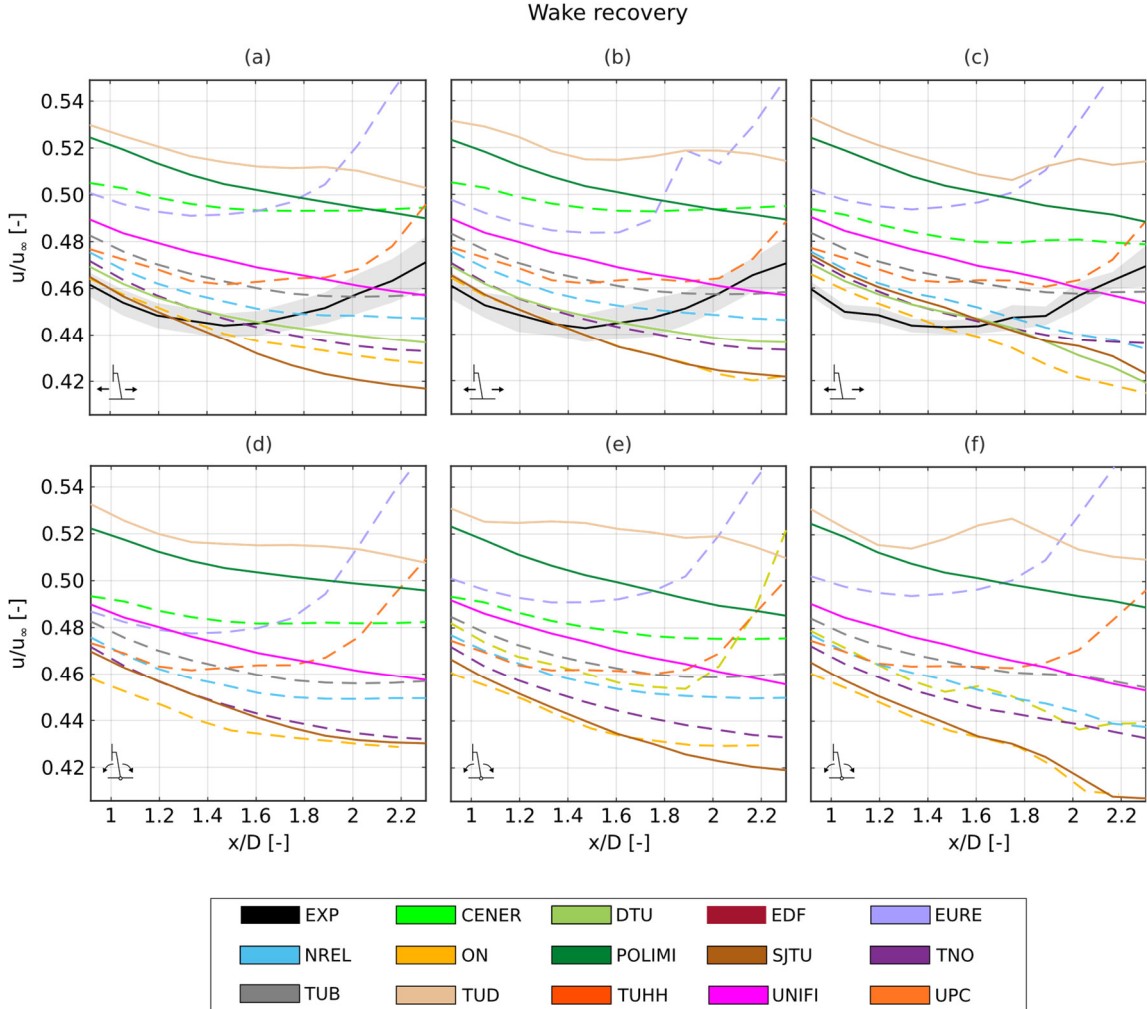

Figure 18 Average streamwise velocity as a function of rotor distance for surge and pitch load cases. The top three figures show the results from the surge cases: (a) LC2.1, $f_r = 0.071$, (b) LC2.5, $f_r = 0.568$, (c) LC2.7, $f_r = 1.137$. The bottom three figures show the results from the pitch cases: (d) LC3.1, $f_r = 0.071$, (e) LC3.5, $f_r = 0.568$, (f) LC3.7, $f_r = 1.137$. Dashed lines represent the FVW results and solid lines represent either the CFD or experimental results. The shaded area represents the standard deviation of the experimental results.

## 9 Conclusions

During the OC6 project, the participants modeled a scaled model of the DTU 10 MW rotor under fixed-bottom, surge, and pitching motion conditions using several numerical approaches with varying levels of fidelity. Simulations were also benchmarked against experimental data obtained during wind tunnel tests by POLIMI. The present study analyzes the characteristics of the near wake (i.e., from 0.25D to 0.5D) and far wake (i.e., from 0.9D to 2.3D) of the rotor, with special focus on the propagation of tip vortices in the near wake.

In the fixed-bottom case, all simulation approaches agree by 0.03D with each other in terms of tip vortex position, and results are consistent with experimental data. The convection velocity calculated from most FVW and CFD simulations falls within one standard deviation of the experimental data, with maximum differences of about 10%. In terms of tip vortex strength, the FVW methods can correctly predict the strength reduction over time; however, differences of up to 20% are observed between the participants. The CFD ALM simulations underpredict the strength to up to 20% despite the increased computational cost. Larger differences were observed in terms of core radius for both FVW and CFD approaches, which indicates some improvements can still be made in predicting this metric. Indeed, for FVW methods this parameter may depend on the blade

discretization used as well as other simulation parameters such as the initial core radius. Careful consideration of these parameters should be carried out when analyzing the tip vortices behavior. In CFD ALM results, the core radius is overpredicted by all the participants, which underlines a limitation of this approach that could also affect its reliability in wake modeling. These results show how correctly tuned FVW methodologies can capture the behavior of the tip vortices at a fraction of the cost of CFD methods.

For unsteady cases, the effect of platform motion on the tip vortex position and strength was evaluated. In surge cases, simulations showed good agreement when the reduced frequency is less than 0.5, with the results oscillating around the steady-state fixed-bottom value as predicted by the linearized quasi-steady theory. However, results diverge from the expected quasi-steady behavior when the reduced frequency increases.

Additionally, the differences across both FVW and CFD approaches increase in those cases, underlining how efforts are probably required to tune the currently available methodologies in order to correctly predict the position and strength of the tip vortices shed from a floating wind turbine under surge motion. Similar conclusions can be drawn when a pitch motion is imposed. Under this condition, however, additional discrepancies between the participants were observed at low frequencies of motion, when the largest tilt angles of the rotor are reached. Additional PIV campaigns are recommended to better understand the evolution of tip vortex strength and position in these conditions. Such a campaign should focus on obtaining measurements at higher reduced frequencies, where wake behavior is yet to be properly characterized. Additionally, the measurements could be performed over multiple cycles of platform motion to provide a more reliable description of the wake response.

Upon examination of hot-wire data, the effect of platform motion on the far wake was further analyzed. Surge and pitch motions induce streamwise velocity oscillations in the wake, which could be problematic for downstream machines, as they could increase rotor loading or affect platform stability. At low frequencies of motion (LC2.1), all simulation methodologies showed that the amplitude of the velocity perturbations in the wake is less than 1% of the free stream value, as is the case for experimental data. However, in LCs 2.7 and 3.7, where the reduced frequency increases up to 1.2, major differences arise between simulation results and experiments. More specifically, most simulations predict a significant increase in the amplitude of velocity oscillations as distance from the rotor increases, even though this trend is not present in the experimental data. Moreover, the codes do not agree on the predicted magnitude of the velocity oscillations. This result suggests that numerical methods might overpredict the effect of platform motion on the wake. This issue affects all simulation methodologies with varying degrees of severity (up to a factor of 3); hence, it does not seem to be connected with a specific approach. A possible cause of the discrepancy between the experimental data and simulations might be unsteadiness (background turbulence and wall effects) in the wind tunnel inflow, which does not allow these instabilities to grow as they do in the simulations. Higher-fidelity CFD models, able to comprehensively model inflow turbulence, are one of the key topics for future aerodynamics studies on FOWTs, although their computational cost is still very high.

Concerning the wake recovery, both simulations and experiment predict a limited effect of surge motion on the velocity recovery at the analyzed frequencies and amplitudes. Additionally, for a pitch motion of the platform, some simulation methodologies showed a slower recovery of the streamwise velocity, even though no experimental data are available to validate this aspect. Overall, current results suggested that the effect of platform motion, differently from original expectations about a faster wake recovery, on the far wake seems to be very limited or even oriented to the generation of a wake less prone to dissipation.

**Appendix A: Characterization of wind tunnel turbulence**

During the experimental campaign the inflow conditions of the wind tunnel were investigated (Fig. A1). Due to the presence of the wind tunnel walls, it was not possible to achieve a perfectly uniform wind speed. However, the velocity was almost

constant from about 0.5m above the wind tunnel floor. The turbulence intensity at the wind tunnel inlet was about 2% above 0.5m, however values up to 10% are measured near to the ground. The integral length scale in the along-wind direction was about 0.2m and was almost constant along the wind tunnel height.

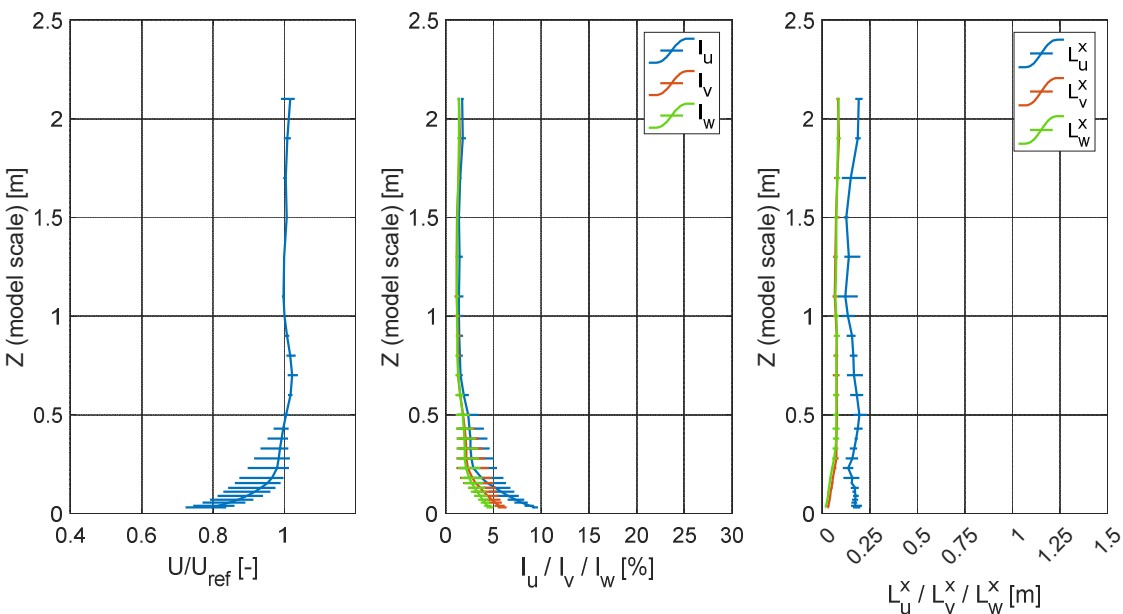

**Figure A1 Characterization of wind tunnel inflow with uncertainty range. (a) Vertical velocity profile normalized by wind speed at**
1005 **hub height, (b) turbulence intensity, (c) integral length scale.**

**Appendix B: Uncertainty of vorticity distribution and tip vortex circulation**

The vorticity calculation requires the spatial derivatives of the velocity field. Because both numerical and experimental data are obtained over a discrete grid, a differentiation scheme is required. For example, considering a central differencing scheme, the velocity derivative at a grid point (i,j) along the x direction is calculated as:

$$\frac{\partial f}{\partial x}(i,j) = \frac{f(i+1,j) - f(i-1,j)}{2\Delta x} + o(\Delta x^2), \qquad (B.1)$$

where $\Delta x$ is the grid spacing and $o(\Delta x^2)$ is the truncation error. The latter is introduced when approximating the flow derivatives using a differentiation scheme, which is usually obtained by performing Taylor's expansion of the velocity field until a finite order. The central differencing scheme is second order accurate, hence the truncation error is of the order of $\Delta x^2$. This error affects both experimental and simulation data. Since the velocity fields are sampled with a resolution of 0.005 m

and 0.007 for the numerical and experimental data, respectively, the truncation error is of the order of $10^{-5}$, and can be considered negligible.

However, for experimental data the calculation of the flow derivatives may also amplify measurement errors that affect the velocity fields obtained with PIV. For this reason, the uncertainty of the vorticity calculation needs to be estimated. For example, following the approach proposed by Sciacchitano and Wieneke (2016), the uncertainty of the vorticity is estimated

as:

$$U_\omega = \frac{U}{\Delta x}\sqrt{1-\rho} \qquad (B.2)$$

where $U_\omega$ and $U$ are the uncertainties of the vorticity and velocity values, respectively, and $\rho$ is the normalized cross-correlation of the measurement error between velocity values at a distance of two grid points. For the PIV measurements performed during the experimental campaign, the maximum in-plane velocity error was estimated as 0.07 m/s, considering a maximum

displacement error of 0.1 pixels. Hence, since the spatial resolution of the data was 0.007 m, the maximum uncertainty of the vorticity value is about 10 1/s, assuming $\rho = 0$. To evaluate the effect of both truncation and measurement errors on the

calculation of the tip vortex strength, the circulation results are checked with the same value calculated as the line integral of the velocity field:

$$\Gamma = \oint \vec{u} \cdot \overrightarrow{dl} \qquad (B.3)$$

The relative error on the tip vortex strength from the two definitions of the circulation is shown in Fig. B1. For both simulations and experiments, the difference between the approaches is less than 1.5%, confirming that estimating the circulation from the vorticity does not introduce significant errors.

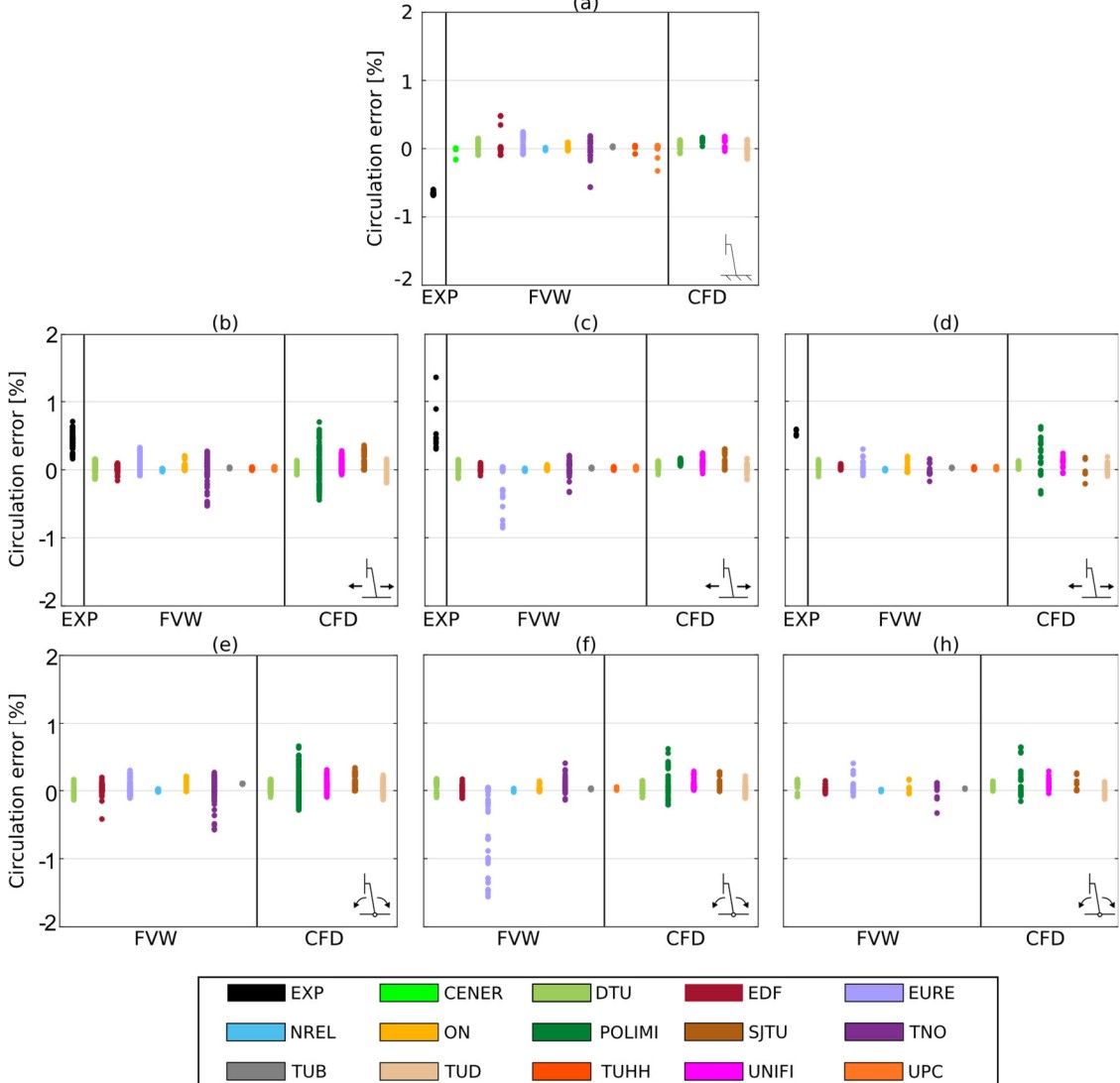

**Figure B1 Relative error of the circulation calculation for both experimental and numerical data. The relative error is estimated by comparing the circulation calculated using Eq. (19) and the value obtained with Eq. (A.3). (a) LC1.1, (b) LC2.1, (c) LC2.5, (d) LC2.7, (e) LC3.1, (f) LC3.5, (g) LC3.7.**

**Supplementary material**

The figures shown in Sects. 7.2.1 and Sect. 8.1 for LC2.5 are provided as supplementary material for all the analysed LCs: 10.5281/zenodo.8210873

**Acknowledgments**

TU Delft would like to acknowledge that this work was carried out with support from the European Industrial Doctorate programme STEP4WIND (H2020-MSCA-ITN-2019, grant agreement 860737). They also acknowledge support from the

Dutch Research Council (NWO) through supercomputing time on the Dutch National Supercomputer Snellius (grant agreement EINF-4659). CENER wants to thank the Government of Navarre for the funding provided by the "Ayudas para la contratación de doctorandos y doctorandas por empresas y organismos de investigación y difusión de conocimientos: doctorados industriales 2019-2021" program that has been used for the development of AeroVIEW. DTU would like to acknowledge that this work was carried out with support from EUDP (case no. 64018-0146). This work was authored in part by the National Renewable Energy Laboratory, operated by the Alliance for Sustainable Energy, LLC, for the US Department of Energy (DOE) under grant no. DEAC36-08GO28308. Funding was provided by the US Department of Energy's offices of Energy Efficiency and Renewable Energy and Wind Energy Technologies. The views expressed in the article do not necessarily represent the views of the DOE or the US Government. The US Government retains and the publisher, by accepting the article for publication, acknowledges that the US Government retains a nonexclusive, paid-up, irrevocable, worldwide license to publish or reproduce the published form of this work, or allow others to do so, for US Government purposes.TUHH kindly thanks the German Federal Ministry for Economic Affairs and Climate Action (BMWK) for funding the IEA Task 30 and the VAMOS project (03EE2004C), which formed a basis for the presented work.

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
