# Peer review of "On the characteristics of the wake of a wind turbine undergoing large motions caused by a floating structure: an insight based on experiments and multi-fidelity simulations from the OC6 Phase III Project"

_Wind Energy Science, 2023_

## Referee Comment (RC2)

**Review of Manuscript wes-2023-21**

**Title**

On the characteristics of the wake of a wind turbine undergoing large motions caused by a floating structure: an insight based on experiments and multi-fidelity simulations from the OC6 Phase III Project

Authors: Cioni, Stefano et al. (more than 30)

**Summary as understood by the referee:**

This manuscript discusses findings from a co-operational project investigating the ability of simulation methods to reproduce the measured wake behaviour of a model wind turbine subjected by "large" movement as are seen, for example in floating off-shore wind-turbines.

This paper is important as it compares mainly vortical methods (assumed to be fully inviscid) and RANS-CFD (includes viscosity driven effects as e.g. turbulence).

**Major:**

Please give a clear definition how "vortex core" is defined (usually where $u_{\phi} \sim r$ ends)

Lines 165 ff:  I do not fully understand of AOA, a etc. refers to the tip only or to a un-specified location on the blade. Please improve.

Line 252: "blockage ratio". Is it an area-based number? Please clarify. If so, 8% is indeed a (too?) large value.

Line 391 (Figure 4) It is a pity that the \omega scales are not identical. Consider to make them equal (min = -100?).

Line 458: (Figure 6 right) consider y-scale to start at 0.7 instead of 0. This would male the differences clearer. Why are there no result from SJTU?

Also, I STRONGLY advice to add ERROR bars to ALL experimental values!  This is very important for quantification of meaning of differences from the various simulations.

Lines 280 ff (section 4). Description of CFD models should include mesh size und turbulence models used and if both were varied to estimate the effect on accuracy

Line 502: Please explain in more detail HOW "the effect of blockage" was considered.

In summary, the authors should be more courageous and draw even stronger conclusions, if possible.

**Minor:**

I wonder if the "large" in the title (and elsewhere) isn't too un-determined.

"OC6". I wonder, if IEAwind Task 30 should be added or a more detailed explanation of the abbreviation.

Line 21: typo: Shangai -> Shanghai

Line 874 (and probably also elsewhere) change "good agreement" to "agreement of xx % or so. As I always say: "good" and "bad" are terms for fairy-tales not from science.

---

## Author Comment (AC1)

**General response to the Reviewers**

Dear reviewers,

We would like to sincerely thank you for your interesting observations that have made improvements in the paper possible.

Based on your comments, we tried our best to improve the paper by clarifying some sections and adding new data and analyses. Modifications have been highlighted in blue-colored text in the revised version of the paper, while a point-to-point response is provided in this document.

We really hope that this revised version can be now worthy of publication in Wind Energy Science.

ᵒᵒᵒ   ᵒᵒᵒ   ᵒᵒᵒ

**Reviewer #1**

1. **In the last paragraph of the introduction: Kleine et al. (2022) did not performed FVW simulations. They used 2 methods: CFD simulations using an actuator line model; and an analytical model based on the Biot-Savart law.**

   Thank you for the right punctuation. Indeed, Kleine et al. performed CFD ALM simulations. The manuscript has been corrected accordingly (line 109).

2. **The call to equation 8 reads "The amplitude of the oscillation can be calculated as Eq. (8)". Equation 8 does not show the amplitude of the oscillation.**

   Equation (8) shows the amplitude of vortex strength oscillation normalized by the amplitude of platform motion. We agree that the clarity of the call to Equation (8) needed clarification, and the text has been modified accordingly (lines 190-191).

3. **The phase is disregarded in equations (8) and (12). However, no justification is included for this. From what I understood, equation (8) is the ratio of maximum change of circulation and maximum tip displacement (analogous for eq. (12)). Please make this explicit. If this is not the case, please clarify.**

   Thank you for the comment, which allowed clarifying an important aspect. The phase shift is in fact disregarded because Eqs. (8) and (12) do represent the extraction of the amplitude from the first order expression of either vortex strength or wake deficit (only shown in the text for the tip vortex strength in Eq. (7)). Indeed, Eq. (7) represents a function of the type $f(x) = \bar{f} + A sin(\omega x + \phi)$, hence the amplitude is given by the factor multiplying the sine, A. This represents half of the maximum oscillation of the investigated metric. Based on your comment, we have tried to further clarify this in the paper (lines 190-191).

4. **In section 2.1, it is worth mentioning that other linearized quasi-steady models might predict a phase shift different from 90°, if different assumptions are used. For example see: Wei, N. J., & Dabiri, J. O. (2022). Phase-averaged dynamics of a periodically surging wind turbine.** *Journal of Renewable and Sustainable Energy*, **14(1), 013305**

   Thank you for your suggestion. The authors agree that it is worth clarifying in the manuscript that other first order models have been proposed which may predict a different aerodynamic response under platform motion. A comment has been added in Section 2.1 (lines 209-216).

5. **Include references for the methods in table 2.**

   References for the methodologies used by the participants have been added (lines 335-339).

6. **Please include the Reynolds number of the experiment (using density and viscosity obtained from local ambient conditions).**

   The Reynolds number was between 20000 and 120000 along the blade span for the test cases analysed in the manuscript (RATED 2 condition), as calculated by Fontanella et al. (https://doi.org/10.5194/wes-6-1169-2021). We have added a Figure below from the reference for completeness. A comment has been added to the

manuscript and the reference to the original publication has been included for the interested reader (Line 267-268).

[Figure]

7. **To allow a better characterization of the inflow conditions, please include more information on the turbulence levels of the wind tunnel. Include at least the integral length scale. If more information is available, please include**.

The Authors have tried their best in adding as many details as possible regarding wind tunnel tests that can aid the comparison of the experimental data with the simulations. In particular, the value of the turbulence intensity was equal to 2% and the integral length scale was about 0.2 m as shown in the attached plot. A comment has been added to the manuscript to provide further information to the reader and the plot below was added as an appendix to the manuscript (Appendix A: Characterization of wind tunnel turbulence). (lines 268-269).

[Figure]

**Figure A1 Characterization of wind tunnel inflow with uncertainty range. (a) Vertical velocity profile normalized by wind speed at hub height, (b) turbulence intensity, (c) integral length scale.**

8.  **Please include the frequency of rotation in Hz. The fact that the frequency of rotation, 4 Hz, is an integer multiple of the platform frequency is a relevant information (this simple calculation should not be left for the reader).**

    The authors agree that reporting the rotational speed in Hz is helpful for the reader. For this reason, the rotational speed is now reported both in Hz and rpm (line 368).

9.  **The phrase "by reducing the amplitude of motion by a factor of 75" in section 5 is not clear. From what I understood, the amplitude of motion, in meters, is reduced for surge but the amplitude of motion, in degrees, is not reduced for pitch. Is that correct? A better way to say is that the non-dimensional amplitude of motion is maintained, where the non-dimensional amplitude of motion is defined as Ax/D for surge and the amplitude angle for pitch**.

    The definition of scaling could indeed be improved. The amplitude of motion is not reduced for a pitch motion of the platform and the scaling was performed by maintaining the non-dimensional amplitude of motion constant. The manuscript has been modified to clarify how the scaling was performed (line 375-376).

10. **My calculations did not agree with the reduced frequency shown in Table 4. For example, for case 2.5, using equation (15): fr=1.0\*2.381/4=0.595, which is different from 0.568. Can you explain?**

    The values of reduced frequency shown in Table 4 are calculated using the corrected velocity of 4.19 m/s which accounts for blockage in the wind tunnel. In order to improve clarity a comment has been added to specify how the reduced frequency was calculated (Caption of Figure 4). Additionally, further details about the calculation of the corrected velocity have been included in the manuscript (lines 276-283).

11. **The circulation calculated using eq. (16) possibly would have significant errors. Please estimate the uncertainties and include error bars in the plots. If this is not possible for all numerical results, please include at least for the experimental results. Please include the "accuracy analysis of the vorticity distributions of both simulations and experiments" as an appendix. If not present in the accuracy analysis, a suggestion is to perform the calculation of the circulation using the line integral of the velocity and compare to area integral of the vorticity (the line integral avoids the derivative).**

    The authors agree that calculating the circulation as the surface integral of the vorticity might introduce errors. For this reason, Appendix A: "Uncertainty of vorticity distribution and tip vortex circulation" has been added to the manuscript.

    Calculating the vorticity requires evaluating the derivatives of the velocity field using a finite differentiation scheme, which introduces a truncation error and could amplify measurement errors in the experimental results.

    The truncation error, using a central differencing scheme is of the order of $\Delta x^2$, where $\Delta x$ is the grid spacing. For numerical and experimental results, the grid spacing is equal to 0.005 m and 0.007 m, respectively, hence the error is of the order of $10^{-5}$ and can be assumed negligible.

    For the experimental results, the amplification of the measurement errors due to the evaluation of the velocity derivatives may not be negligible. In order to provide an approximate estimate of the amplification of errors in the experimental data, the propagation of the velocity error from the PIV measurements to the vorticity calculation was estimated following the methodology described in Sciacchitano and Wieneke (2016). The velocity error from PIV was estimated during the experimental campaign as 0.07 m/s. Since the spatial resolution of the experimental data was 0.007 m and assuming a conservative value of 0 for the correlation factor, the error on the vorticity is estimated as 10 $1/s$.

    In order to evaluate the effect of the uncertainty propagation on the calculation of the vortex strength, the results obtained by integrating the vorticity were compared with the same results obtained by performing the line integral of the velocity. Results showed that the difference between the two methods is smaller than 1.5% in all the analyzed cases (lines 432-436).

12. **Please estimate the uncertainties of the position of the vortex core and include error bars in the plots. If this is not possible for all numerical results, please include at least for the experimental results.**

    The authors agree that including error bars for the streamwise position and core radius of the tip vortex is a valuable addition to the manuscript.

    For experimental data, error bars have been included for the streamwise position, convection velocity, core radius and vortex strength in the fixed-bottom case (Figures 6 and 7 and lines 461-466). This was possible as during

the experimental tests the velocity fields were obtained by averaging the PIV data over 100 rotor revolutions. Hence, the raw, non-averaged data, was postprocessed in order to determine the standard deviation of these metrics, providing insight into the uncertainty of these values.

In the surge cases, the experimental dataset only includes the velocity fields from a single surge cycle. Hence, it was not possible to estimate the standard deviation of the tip vortex metrics, either in terms of average value or in terms of amplitude and phase shift. This remains one of the limitations of this work. However, the authors believe that the inclusion of the available experimental data provides valuable insight into the analysis, despite the lack of error bars. A comment has been added to the text to underline this limitation of the current study (lines 466 and 576-578).

For the numerical data, the participants ran the simulations over multiple revolutions and cycles of platform motion in order to achieve convergence of the result. Hence, no significant differences are expected over multiple cycles. A comment has been added to the manuscript to clarify this point (line 319-321).

13. **The phrase "The vorticity plots are shown using a threshold of \omega = 5 m^2/s" is not clear in the legend of figure 4. The threshold does not seem to be 5.**

Thank you for the right suggestion. The threshold used to plot the results is now shown in the legend. Results have been double checked to ensure that the threshold is 5 $m^2/s$ and the color bar edges have been modified as suggested by the second reviewer.

14. **In equation (18): is * a symbol for multiplication?**

The symbol "*" stands for multiplication in Equation (18). After reviewing the manuscript, the symbol was modified to " · " to maintain consistency with other equations in the text and avoid possible confusion.

15. **Regarding phase shift for the experimental results: Do I understand correctly: only 4 points are used to calculate the phase shift experimentally? It seems to me that 4 points is too low for experimental data. My perception is that a small uncertainty or noise in the value of the function at the points would lead to very different results of phase shift. Also, I have the impression that the error related to aliasing would be very significant. Please investigate this question in detail. Before comparing the numerical results to the experimental phase shift, the authors should show that the errors are low. If the experimental phase shift results are not reliable, the discussion should be revised.**

It is confirmed that the amplitude and phase shift of the experimental data for LC2.5 were calculated using the only 4 available points, as they are shown in Figures 8 and 9, due to the lack of available experimental data. For LC2.1, more data points are available due to the slower surge motion, hence the prediction of the amplitude and phase shift is more reliable. The authors agree that calculating the amplitude and phase shift from only four points may result in large errors due to aliasing. However, experimental results were included in the manuscript in order to provide an estimate from the available experimental data. Nevertheless, the authors agree that the reliability of the experiments should be clarified in the manuscript and a comment has been added in Sect. 7.2.1 to underline this limitation (line 586-588).

16. **It would be very useful if the data used to calculate the quantities in section 7.2.2 could be provided as supplementary material (or in an appendix) in the format of the plots of section 7.2.1. The plots of section 7.2.1 give important information to interpret the results of section 7.2.2. For example, it is possible to observe in figure 9 that the method used to calculate the vortex strength does not seem very reliable for some methods (example: SJTU and NREL).**

The authors agree that the additional plots may provide useful information for the interested reader. The plots of Section 7.2.1 have been added as supplementary material for all the LCs analyzed in the manuscript. The supplementary material is added at the end of this rebuttal and the additional plots will be published to an open access repository on zenodo after publication at the link: 10.5281/zenodo.8210873. We have added the Section "Supplementary material" reporting the link to the additional plots and included additional comments in the manuscript (lines 578-579 and 843-844).

17. **Results in general: please show the distance in the streamwise direction and the amplitudes in non-dimensional format (x/D and A/D).**

Figures 6, 7, 11, 14, 15, 17, and 18 have been modified by normalizing the streamwise positions and the amplitudes of motion.

18. **Suggestion: show the velocities in the streamwise direction (u/Uinf) and other quantities in non-dimensional format.**

The results for the streamwise position of the tip vortex and streamwise velocity have been normalized in the respective Figures.

**Some typos and other presentation comments:**

1. **Missing "et al." in some references (examples: Arabgolarcheh (2022) and Ramos-Garcia (2022a)).**

Thank you for your comment. We have corrected the references in the manuscript (lines 96 and 98)

2. **Equation (14): \Omega does not follow nomenclature of this paper.**

Thank you for your comment. Equation (14) has been modified to follow the nomenclature used in this paper.

3. **In section 7.2.2: reference to figure 10, instead of figure 11.**

The reference has been corrected (line 696).

4. **References: many references included as pre-prints have already been published.**

Thank you for your comment. We have checked the references included in the manuscript and corrected them.

**Other Changes:**

Due to a post-processing issue Figures 11, 12 and 13 have been modified, resulting in some small differences for some of the participants. The text has been modified to reflect these changes.

**Supplementary material**

In this file the plots that were not shown in the article are reported here. For all the plots, experimental data are shown in black and for each participant a specific color is assigned as shown in Figure 1.

[Figure]

**Figure 1 List of particpants and corresponding color.**

1. **Tip vortex metrics**

In this Section the plots concerning the tip vortex metrics for all the LCs analysed during this work are reported.

**1.1. Tip vortex streamwise position**

[Figure]

**Figure 2 Streamwise position of the tip vortex during a cycle of platform motion for LC2.1 at a vortex age of 409°. Dashed lines represent the FVW results, solid lines represent the CFD results, and the black crosses represent the experimental data. Red diamonds indicate the fixed-bottom value.**

[Figure]

**Figure 3 Streamwise position of the tip vortex during a cycle of platform motion for LC2.5 at a vortex age of 427°. Dashed lines represent the FVW results, solid lines represent the CFD results, and the black crosses represent the experimental data. Red diamonds indicate the fixed-bottom value.**

[Figure]

**Figure 4 Streamwise position of the tip vortex during a cycle of platform motion for LC2.7 at a vortex age of 408°. Dashed lines represent the FVW results, solid lines represent the CFD results, and the black crosses represent the experimental data. Red diamonds indicate the fixed-bottom value.**

[Figure]

**Figure 5 Streamwise position of the tip vortex during a cycle of platform motion for LC3.1 at a vortex age of 409°. Dashed lines represent the FVW results and solid lines represent the CFD results. Red diamonds indicate the fixed-bottom value.**

[Figure]

**Figure 6 Streamwise position of the tip vortex during a cycle of platform motion for LC3.5 at a vortex age of 427°. Dashed lines represent the FVW results and solid lines represent the CFD results. Red diamonds indicate the fixed-bottom value.**

[Figure]

**Figure 7 Streamwise position of the tip vortex during a cycle of platform motion for LC3.7 at a vortex age of 408°. Dashed lines represent the FVW results and solid lines represent the CFD results. Red diamonds indicate the fixed-bottom value.**

**1.2 Tip vortex strength**

[Figure]

**Figure 8** Tip vortex strength during a cycle of platform motion for LC2.1 at a vortex age of 409°. Dashed lines represent the FVW results, solid lines represent the CFD results, and the black crosses represent the experimental data. Red diamonds indicate the fixed-bottom value.

[Figure]

**Figure 9** Tip vortex strength during a cycle of platform motion for LC2.5 at a vortex age of 427°. Dashed lines represent the FVW results, solid lines represent the CFD results, and the black crosses represent the experimental data. Red diamonds indicate the fixed-bottom value.

[Figure]

**Figure 10 Tip vortex strength during a cycle of platform motion for LC2.7 at a vortex age of 408°. Dashed lines represent the FVW results, solid lines represent the CFD results, and the black crosses represent the experimental data. Red diamonds indicate the fixed-bottom value.**

[Figure]

**Figure 11 Tip vortex strength during a cycle of platform motion for LC3.1 at a vortex age of 409°. Dashed lines represent the FVW results and solid lines represent the CFD results. Red diamonds indicate the fixed-bottom value.**

[Figure]

**Figure 12 Tip vortex strength during a cycle of platform motion for LC3.5 at a vortex age of 427°. Dashed lines represent the FVW results and solid lines represent the CFD results. Red diamonds indicate the fixed-bottom value.**

[Figure]

**Figure 13 Tip vortex strength during a cycle of platform motion for LC3.7 at a vortex age of 408°. Dashed lines represent the FVW results and solid lines represent the CFD results. Red diamonds indicate the fixed-bottom value.**

**1.3 Tip vortex core radius**

[Figure]

[Figure]

**Figure 14 (left)Tip vortex core radius during a cycle of platform motion for LC2.1 at a vortex age of 409°. Dashed lines represent the FVW results, solid lines represent the CFD results, and the black crosses represent the experimental data. (right) Average core radius during a cycle of motion. The whiskers represent the minimum and maximum values of the core radius during the cycle of motion, and the red diamonds indicate the fixed-bottom value.**

[Figure]

[Figure]

**Figure 15 (left)Tip vortex core radius during a cycle of platform motion for LC2.5 at a vortex age of 427°. Dashed lines represent the FVW results and solid lines represent either the CFD results or the experimental data. (right) Average core radius during a cycle of motion. The whiskers represent the minimum and maximum values of the core radius during the cycle of motion, and the red diamonds indicate the fixed-bottom value.**

[Figure]

**Figure 16 (left)**Tip vortex core radius during a cycle of platform motion for LC2.7 at a vortex age of 408°. Dashed lines represent the FVW results and solid lines represent either the CFD results or the experimental data. **(right)** Average core radius during a cycle of motion. The whiskers represent the minimum and maximum values of the core radius during the cycle of motion, and the red diamonds indicate the fixed-bottom value.

[Figure]

**Figure 17 (left)**Tip vortex core radius during a cycle of platform motion for LC3.1 at a vortex age of 409°. Dashed lines represent the FVW results and solid lines represent the CFD results. **(right)** Average core radius during a cycle of motion. The whiskers represent the minimum and maximum values of the core radius during the cycle of motion, and the red diamonds indicate the fixed-bottom value.

[Figure]

**Figure 18 (left)Tip vortex core radius during a cycle of platform motion for LC3.5 at a vortex age of 427°. Dashed lines represent the FVW results and solid lines represent the CFD results (right) Average core radius during a cycle of motion. The whiskers represent the minimum and maximum values of the core radius during the cycle of motion, and the red diamonds indicate the fixed-bottom value.**

[Figure]

**Figure 19 (left)Tip vortex core radius during a cycle of platform motion for LC3.7 at a vortex age of 408°. Dashed lines represent the FVW results and solid lines represent the CFD results. (right) Average core radius during a cycle of motion. The whiskers represent the minimum and maximum values of the core radius during the cycle of motion, and the red diamonds indicate the fixed-bottom value.**

**2 Hot wire data**

In this Section the plots concerning the hot wire data for all the LCs analysed during this work are reported.

**2.1 Wake deficit**

[Figure]

**Figure 20 (left) Wake deficit oscillations during platform motion for LC2.1. Dashed lines represent the FVW results and solid lines represent either the CFD or experimental results. The shaded area represents the standard deviation of the experimental data (right) Amplitude of the wake deficit oscillations at the frequency of platform motion. The red lines represent the average wake deficit during the cycle of motion, and the red diamonds represent the fixed-bottom value.**

[Figure]

**Figure 21 (left) Wake deficit oscillations during platform motion for LC2.5. Dashed lines represent the FVW results and solid lines represent either the CFD or experimental results. The shaded area represents the standard deviation of the experimental data (right) Amplitude of the wake deficit oscillations at the frequency of platform motion. The red lines represent the average wake deficit during the cycle of motion, and the red diamonds represent the fixed-bottom value.**

[Figure]

**Figure 22 (left) Wake deficit oscillations during platform motion for LC2.7. Dashed lines represent the FVW results and solid lines represent either the CFD or experimental results. The shaded area represents the standard deviation of the experimental data (right) Amplitude of the wake deficit oscillations at the frequency of platform motion. The red lines represent the average wake deficit during the cycle of motion, and the red diamonds represent the fixed-bottom value.**

[Figure]

**Figure 23 (left) Wake deficit oscillations during platform motion for LC3.1. Dashed lines represent the FVW results and solid lines represent the CFD results. (right) Amplitude of the wake deficit oscillations at the frequency of platform motion. The red lines represent the average wake deficit during the cycle of motion, and the red diamonds represent the fixed-bottom value.**

[Figure]

**Figure 24 (left) Wake deficit oscillations during platform motion for LC3.5. Dashed lines represent the FVW results and solid lines represent the CFD results. (right) Amplitude of the wake deficit oscillations at the frequency of platform motion. The red lines represent the average wake deficit during the cycle of motion, and the red diamonds represent the fixed-bottom value.**

[Figure]

**Figure 25 (left) Wake deficit oscillations during platform motion for LC3.7. Dashed lines represent the FVW results and solid lines represent the CFD results. (right) Amplitude of the wake deficit oscillations at the frequency of platform motion. The red lines represent the average wake deficit during the cycle of motion, and the red diamonds represent the fixed-bottom value.**

**2.2 Streamwise velocity oscillations**

[Figure]

**Figure 26 (left) Streamwise velocity oscillations during platform motion from a single HWA probe at x = 5.48 m and y = 0.9 m for LC2.1. Dashed lines represent the FVW results and solid lines represent either the CFD or experimental results. The shaded area represents the standard deviation of the experimental data (right) Amplitude of the streamwise velocity oscillations calculated at the frequency of platform motion. The red lines represent the average values during the cycle of motion, and the red diamonds represent the fixed-bottom case.**

[Figure]

**Figure 27 Streamwise velocity oscillations during platform motion from a single HWA probe at x = 5.48 m and y = 0.9 m for LC2.5. Dashed lines represent the FVW results and solid lines represent either the CFD or experimental results. The shaded area represents the standard deviation of the experimental data (right) Amplitude of the streamwise velocity oscillations calculated at the frequency of platform motion. The red lines represent the average values during the cycle of motion, and the red diamonds represent the fixed-bottom case.**

[Figure]

**Figure 28 Streamwise velocity oscillations during platform motion from a single HWA probe at x = 5.48 m and y = 0.9 m for LC2.7. Dashed lines represent the FVW results and solid lines represent either the CFD or experimental results. The shaded area represents the standard deviation of the experimental data (right) Amplitude of the streamwise velocity oscillations calculated at the frequency of platform motion. The red lines represent the average values during the cycle of motion, and the red diamonds represent the fixed-bottom case.**

[Figure]

**Figure 29 Streamwise velocity oscillations during platform motion from a single HWA probe at x = 5.48 m and y = 0.9 m for LC3.1. Dashed lines represent the FVW results and solid lines represent the CFD results. (right) Amplitude of the streamwise velocity oscillations calculated at the frequency of platform motion. The red lines represent the average values during the cycle of motion, and the red diamonds represent the fixed-bottom case.**

[Figure]

**Figure 30 Streamwise velocity oscillations during platform motion from a single HWA probe at x = 5.48 m and y = 0.9 m for LC3.5. Dashed lines represent the FVW results and solid lines represent the CFD results. (right) Amplitude of the streamwise velocity oscillations calculated at the frequency of platform motion. The red lines represent the average values during the cycle of motion, and the red diamonds represent the fixed-bottom case.**

[Figure]

**Figure 31 Streamwise velocity oscillations during platform motion from a single HWA probe at x = 5.48 m and y = 0.9 m for LC3.7. Dashed lines represent the FVW results and solid lines represent the CFD results. (right) Amplitude of the streamwise velocity oscillations calculated at the frequency of platform motion. The red lines represent the average values during the cycle of motion, and the red diamonds represent the fixed-bottom case.**

---

## Author Comment (AC2)

**General response to the Reviewers**

Dear reviewers,

We would like to sincerely thank you for your interesting observations that have made improvements in the paper possible.

Based on your comments, we tried our best to improve the paper by clarifying some sections and adding new data and analyses. Modifications have been highlighted in blue-colored text in the revised version of the paper, while a point-to-point response is provided in this document.

We really hope that this revised version can be now worthy of publication in Wind Energy Science.

ooo  ooo  ooo

**Reviewer #2**

**1. Please give a clear definition how "vortex core" is defined.**

Thank you for this rightful comment. The vortex core is defined as the inner part of a vortex, where the fluid rotates as a rigid body. This region is characterized by high vorticity values. The vortex core radius is defined in this work as the distance between the velocity maximum of the swirling velocity, following the methodology commonly used in the literature (see for example, van der Wall et al. DOI: 10.1007/s00348-006-0117-x). We have clarified in the manuscript the definition of the vortex core (lines 411-413).

**2. Lines 165: I do not fully understand of AOA, a etc. refers to the tip only or to a unspecified location on the blade. Please improve.**

The authors agree that this part of the manuscript could be improved. In the manuscript the circulation is assumed to be a function of the flow incidence in the last portion of the blade, which is in turn related to the angle of attack at the tip of the blade. The manuscript has been modified to clarify this point (line 171).

**3. Line 252: "blockage ratio". Is it an area-based number? Please clarify. If so, 8% is indeed a (too?) large value.**

The authors agree that the definition of the blockage ratio in the manuscript could be improved. The blockage ratio was calculated as the area ratio, $\beta = A_{rot}/A_{test\ section}$. The definition of the blockage ratio has been clarified in the manuscript (lines 272-274, and Eq. (16)).

We agree that the value of 8% is significant. For this reason the simulations that did not include the wind tunnel walls used a corrected inflow velocity equal to 4.19 m/s to account for the acceleration of the flow in the wind tunnel (lines 276-283).

**4. Line 391 (Figure 4) It is a pity that the \omega scales are not identical. Consider to make them equal (min = -100?).**

The authors agree that modifying the scales can improve the visualization of the results. Figure 4 was modified accordingly, by changing the scale to 0/-100.

**5. Line 458: (Figure 6 right) consider y-scale to start at 0.7 instead of 0. This would male the differences clearer. Why are there no result from SJTU?**

The authors agree that modifying the scale to start from 0.7 improves the clarity of the plot. Figure 6 has been modified accordingly. For the fixed-bottom case (LC1.1) no data from SJTU was available, hence it was not possible to include results in the corresponding figures.

**6. Also, I STRONGLY advice to add ERROR bars to ALL experimental values! This is very important for quantification of meaning of differences from the various simulations.**

The authors agree that adding error bars to the experimental data is valuable to the present work. For PIV data, error bars have been included for all the fixed-bottom cases, as the velocity fields were acquired for 100 rotor revolutions.

A comment has been added in the methodology about how the standard deviation of the experimental data was evaluated (lines 461-466). Hence, the standard deviation of all the investigated metrics was included in the plots. However, it was not possible to include error bars for the surge cases, as the velocity fields were acquired for a single cycle of surge motions. The authors agree that this is a limitation of the current study. For this reason, this point was emphasized in the manuscript to clarify that further experimental tests are required to validate the initial results shown in this work (lines 576-579).

Regarding hotwire data, the standard deviation of the streamwise velocity and the wake deficit was added in the figures. However, it was not possible to estimate the uncertainty of the amplitudes and phase-shifts due to the limited number of experimental data available. A comment has been added to the manuscript to underline this limitation for the reader (lines 488-491 and 900-901).

**7. Lines 280 ff (section 4). Description of CFD models should include mesh size und turbulence models used and if both were varied to estimate the effect on accuracy**

The authors agree that further details about the CFD models could improve the description of the methodologies employed by the participants. Further details have been included in the manuscript and summarized in Table 4. The participants tested different mesh-sizes in order to evaluate the effect of the grid sizing on the results and to guarantee accuracy. Instead, the effect of turbulence models on the results was not evaluated. A comment has been added to the manuscript to clarify this point (lines 350-351 and 344-345).

**Table 1 Main simulation parameters for CFD simulations**

| Participant | POLIMI | SJTU | TUD | UNIFI |
|---|---|---|---|---|
| Simulation approach | ALM URANS | Blade resolved DES | ALM LES | ALM URANS |
| Turbulence model | k-$\omega$ SST | Spalart-Allmaras | Dynamic Smagorinsky | k-$\varepsilon$ RNG |
| Rotor region [D] | 0.26 | 0.11 | 2.5 | 0.22 |
| Rotor region cell size | $1.7 \cdot 10^{-2} m$ | $3 \cdot 10^{-3} m$ | $1.3 - 2.4 \cdot 10^{-2} m$ | $1.56 \cdot 10^{-2} m$ |
| Near wake region [D] | 0.63 | 6.26 | N.A. | 0.84 |
| Near wake element size | $2 \cdot 10^{-2} m$ | $1.2 \cdot 10^{-2} m$ | N.A. | $3.13 \cdot 10^{-2} m$ |
| Far wake element size | $4.5 \cdot 10^{-2} m$ | $4.8 \cdot 10^{-2} m$ | $5 \cdot 10^{-2}$ | $6.25 \cdot 10^{-2} m$ |

**8. Line 502: Please explain in more detail HOW "the effect of blockage" was considered.**

The authors agree that the correction of blockage in the simulations could be improved in the manuscript. Additional comments have been added to the manuscript (lines 276-283). Since the wind tunnel blockage affects the results, the simulations run by POLIMI, TUD and UNIFI included the wind tunnel walls. The remaining participants, which could not include the walls in their simulations, corrected the free stream velocity in order to account for the flow acceleration in the wind tunnel. The corrected free stream velocity was calculated using the correction proposed by Glauert for moderate blockage ratios,

$$U_{\infty}' = U_{\infty} \left( 1 + \frac{\beta C_t}{4\sqrt{1 - C_t}} \right)^{-1}$$

where $\beta$ is the area-based blockage ratio, and $U_{\infty}'$ and $U_{\infty}$ are the corrected and actual free-stream velocities. The parameter $C_t$ is the thrust coefficient, calculated as:

$$C_t = \frac{T}{0.5\rho A_d U_{\infty}^2}$$

For the present case $C_t$ is about 0.88, $U_\infty$ is 4m/s and the air density is 1.177 kg/m³ resulting in a corrected wind speed of 4.19 m/s.

**9. In summary, the authors should be more courageous and draw even stronger conclusions, if possible.**

From the analysis of the numerical and experimental results no clear trend has been identified that suggests a consistent limitation of one of the methodologies employed. Moreover, it has to be remembered that current data are unique but limited in terms of acquisition length. The numerical results show better agreement for the fixed-bottom case than in the unsteady cases, where significant differences arise among the participants especially at high frequencies of motion. This suggests that the currently available methods require further tuning in order to capture the wake behavior in these conditions as different methodologies could show significant discrepancies and lead to different conclusions concerning the wake response of a floating wind turbine. Further analysis and especially experimental tests are required to identify the sources of the observed limitations and to validate the currently available results. We have modified the conclusions to further clarify these points.

**Minor**

**10. I wonder if the "large" in the title (and elsewhere) isn't too un-determined.**

The reviewer is right. Of course, it is difficult to indicate thresholds in these applications but the inclusion of "large" in the title is intended to emphasize that the investigated amplitudes of motion correspond to significant ones when translated to full scale. Indeed, for the surge motion, the maximum amplitude is 0.125m which corresponds to 9.375m at full scale and a total displacement in the wind direction of 18.75m. For the pitch motion an amplitude of 3° is considered corresponding to a maximum displacement of 6°.

**11. "OC6". I wonder, if IEAwind Task 30 should be added or a more detailed explanation of the abbreviation.**

The manuscript has been modified to explain the abbreviation and specify that the project was carried out under the IEA wind Task 30 (lines 115-116).

**12. Line 21: typo: Shangai -> Shanghai**

Thank you for your comment. We have corrected the manuscript (line314).

**13. Line 874 (and probably also elsewhere) change "good agreement" to "agreement of xx % or so. As I always say: "good" and "bad" are terms for fairy-tales not from science.**

Thank you for your comment. The manuscript has been corrected to provide a better quantification of the results, wherever possible.

**Other Changes:**

Due to a post-processing issue Figures 11, 12 and 13 have been modified, resulting in some small differences for some of the participants. The text has been modified to reflect these changes.

---

## Author Response (AR2)

Firenze, 20/09/2023

Dear Editor,

thank you for the possibility of further improving our study. We have included all the suggestions in this revised version. Modifications have been highlighted in blue-colored text in the revised version of the paper.

We really hope that this revised version can be now worthy of publication in Wind Energy Science.

Best regards,

*Alessandro Bianchini*

on behalf of all the authors

---

## Author Response (AR3)

Firenze, 28/09/2023

Dear Prof. Aubrun,

indeed, we already addressed the comments of the Reviewer in the revised version of the paper. Main changes were made at lines 513-518 and 532-533 (plus some small additional corrections). We forgot to mention the lines in the rebuttal and to highlight them in the paper (we only used blue-colored text), and I think this may have made them not sufficiently visible. Sorry for this.

Thank you for your management of the review process.

Best regards,

*Alessandro Bianchini*

on behalf of all the authors